# Distinct waves of ovarian follicles contribute to mouse oocyte production

Qi Yin, Allan C Spradling*

Howard Hughes Medical Institute and Department of Biosphere Sciences, Carnegie Institution for Science, Baltimore, United States

## eLife Assessment

This **important** study reports that two distinct waves of ovarian follicles contribute to oocyte production in mice. The paper provides large amounts of data that will benefit future studies, although the methods and analysis are considered **incomplete** at present. Justification for the criteria of wave 1 follicles would benefit from further explanation and discussion. This work will be of interest to ovarian biologists and physicians working on female infertility.

**Abstract** The earliest growing mouse follicles, wave 1, rapidly develop in the ovarian medulla, while the great majority, wave 2, are stored for later use as resting primordial follicles in the cortex. Wave 1 follicles are known to mostly undergo atresia, a fate sometimes associated with the persistence of steroidogenic theca cells, but this connection is poorly understood. We characterized wave 1 follicle biology using tissue clearing, lineage tracing, and scRNA-seq to clarify their contributions to offspring and steroidogenesis. Wave 1 follicles, lineage-marked by E16.5 *Foxl2* expression in granulosa cells, reach preantral stages containing theca cell layers by 2 weeks. Atresia begins about a week later, during which 80–100% of wave 1 follicles degrade their oocytes, turn over most granulosa cells, but retain theca cells which expand in number together with interstitial gland cells in the medulla. During puberty (5 weeks), these cells ultrastructurally resemble steroidogenic cells and highly express androgen biosynthetic genes. Unexpectedly, the *Foxl2* lineage tag also marked about 400 primordial follicles, located near the medullary–cortical boundary, that become the earliest activated wave 2 follicles. These 'boundary' or 'wave 1.5' follicles generate 70–100% of the earliest mature oocytes, while fewer than 26 wave 1 follicles with oocytes survive. Consistent with their largely distinct fates in steroid or oocyte production, granulosa cells of antral wave 1 and 2 follicles differentially express multiple genes, including *Wnt4* and *Igfbp5*.

*For correspondence: spradling@carnegiescience.edu

Competing interest: The authors declare that no competing interests exist.

## Introduction

Gametes often begin development in subgroups known as 'waves' that arise at specific times and gonadal locations (*Hirshfield, 1992*; *Hirshfield and DeSanti, 1995*; *Gougeon, 1996*; *Jiménez, 2009*; *DeFalco and Capel, 2009*; *Zheng et al., 2014a*; *Zheng et al., 2014b*). In mice, the earliest developing follicles, a small group known as 'wave 1', become growing primary follicles in the ovarian medulla by the time of birth without entering dormancy as primordial follicles. The remaining ~90%, 'wave 2', enter quiescence before postnatal day 5 (P5) by forming primordial follicles within the ovarian cortex to support lifelong fertility (*Edson et al., 2009*; *McKey et al., 2022*; *Figure 1A*). In the medulla, wave 1 follicles retain early-arising bipotential pre-granulosa (BPG) cells, which express *Foxl2* and *Nr5a2* beginning during early fetal development. In contrast, wave 2 follicles in the cortex acquire epithelial pre-granulosa (EPG) cells, express *Foxl2* only after birth, and mostly express little *Nr5a2* as primordial follicles (*Mork et al., 2012*; *Zheng et al., 2014a*; *Niu and Spradling, 2020*; *Meinsohn et al., 2021a*).

However, a small subset of non-growing primordial follicles was identified that did express *Nr5a2* and were proposed to be primed for activation (*Meinsohn et al., 2021b*).

Once primordial follicles are signaled to grow, their granulosa cells proliferate and soon form multiple granulosa cell layers surrounding the oocyte (*Habara et al., 2021*). Granulosa cells in recruited follicles secrete signals encoded by Desert hedgehog (*Dhh*) and Indian hedgehog (*Ihh*) that attract ovarian mesodermal precursors to form outer theca cell layers on follicles and to synthesize steroid hormones (*Liu et al., 2015*). Additionally, ovarian medullary precursors generate thecal fibroblastic cells and perivascular smooth muscle cells that vascularize the thecal region and enable these follicles to collectively serve as a major systemic steroid hormone source (*Guzmán et al., 2023*). An ovarian cell type closely related to theca cells, interstitial gland cells, also coordinately forms in mice and other mammals (*Jiménez, 2009*; *Miyabayashi et al., 2015*). *Nr5a2* promotes theca cell vascularization that is essential for ovulation (*Duggavathi et al., 2008*; *Bianco et al., 2019*; *Guzmán et al., 2022*). Two classes of steroidogenic Leydig cells, fetal and adult, also arise during testis development (*Miyabayashi et al., 2015*; *Liu et al., 2016*).

Follicular cells synthesize specific steroid hormones during their development (*Edson et al., 2009*; *Richards et al., 2018*; *Niu and Spradling, 2020*; *Meinsohn et al., 2021a*). Steroidogenic theca cells of secondary and preantral follicles convert cholesterol into androgens through a series of enzymatic reactions involving steroidogenic acute regulatory protein (StAR), cytochrome P450 family 11 subfamily A member 1 (CYP11A1), hydroxy-delta-5-steroid dehydrogenase, 3-beta- and steroid delta-isomerase 1 (HSD3B1), cytochrome P450 family 17 subfamily A member 1 (CYP17A1), and steroid 5 alpha-reductase 1 (SRD5A1). *Cyp17a1-iCre* (*Cyp17a1* promoter driving codon-improved *Cre*) has been developed and used to label theca cells, interstitial gland cells, and some granulosa cells (*Bridges et al., 2008*). Ovarian androgen production by theca cells during juvenile development is essential for normal maturation and the onset of puberty (*Galas et al., 2012*; *Walters et al., 2019*; *Kelava et al., 2022*). In addition, androgens diffuse into neighboring granulosa cells, where they are converted into estrogen by the action of hydroxysteroid 17-beta dehydrogenase 1 (HSD17B1) and cytochrome P450 family 19 subfamily A member 1 (CYP19A1). Estrogen produced in this manner signals the hypothalamus to initiate gonadotropin production and is essential for subsequent follicular maturation. After ovulation, the oocyte-free follicle is further modified into a corpus luteum, and its remaining granulosa and theca cells undergo luteinization to generate progesterone.

The development and function of wave 1 follicles have remained controversial. Most wave 1 follicles undergo atresia after reaching early antral stages (*Byskov, 1974*; *Erickson et al., 1985*) during juvenile development (*Figure 1B*). This suggests they do not primarily serve a direct reproductive function (*Hirshfield, 1992*; *Hsueh et al., 1994*; *Eppig and Handel, 2012*), although these follicles are capable of developing into functional oocytes following superovulation (*Lamas et al., 2021*) and can support fertility in an irradiated rat model (*Guigon et al., 2003*). Theca cells of atretic follicles often survive and persist in the ovary (*Erickson et al., 1985*; *Magoffin, 2005*), suggesting a role in steroid production. However, when *Foxl2-CreERT2* was used to lineage-label the granulosa cells of wave 1 follicles, marked follicles substantially contributed to offspring during the initial months of fertility (*Zheng et al., 2014b*).

Here, we studied wave 1 follicle development and gene expression in detail to gain deeper insights into follicular waves. We found that lineage labeling using *Foxl2-CreERT2*, activated in early fetal stages, is not specific to wave 1 follicles. Granulosa cells in about 400 primordial follicles were also marked near the medullary–cortical boundary in 2-week-old mice; hence, we termed them 'boundary' or 'wave 1.5' follicles. Marked boundary follicles accounted for more than 70% of developing follicles in the 5 week ovary, while no more than 26 intact wave 1 follicles survived to puberty. Boundary follicles probably correspond to the resting primordial follicle population expressing *Nr5a2* described by *Meinsohn et al., 2021a*; *Meinsohn et al., 2021b*. Thus, most of the *Foxl2-CreERT2*-labeled follicles contributing to offspring observed by *Zheng et al., 2014a* likely derived from boundary follicles rather than from wave 1 follicles.

Ovarian developmental and gene expression studies during 2–6 weeks (wk) provided molecular insight into gamete waves and follicular atresia. Wave 1 follicular oocytes and granulosa cells largely turn over beginning at about 3 wk. However, their theca cells increase in number and, by puberty, join interstitial gland cells to form a mass of active steroidogenic medullary cells that retain morphological evidence of their origin from atretic follicles. These steroidogenic cells highly express androgenic

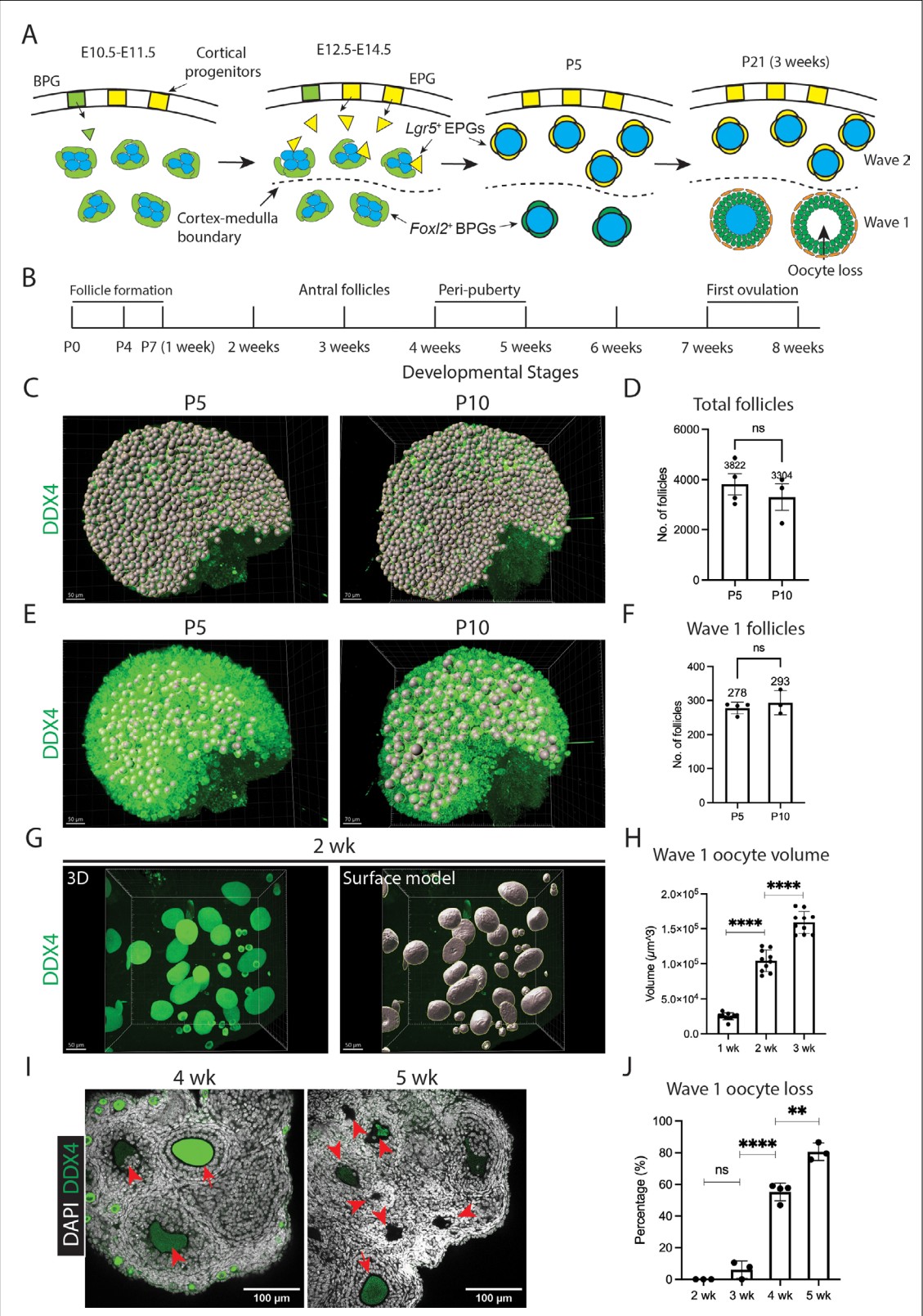

**Figure 1.** Wave 1 follicles are remodeled prior to peri-puberty. (**A**) Schematic diagrams of fetal and postnatal ovaries at the indicated stages (E = embryonic day; P = postnatal day). Wave 1 follicles, located in the ovarian medulla (below the dashed line), retain bipotential pre-granulosa (BPG, green) cells and develop without delay. In contrast, cortical follicles, which become quiescent primordial follicles (wave 2), undergo replacement of BPGs by epithelial pre-granulosa (EPG, yellow) cells. This replacement is completed by P5. By P21 (3 wk), wave 1 follicles undergo atresia, characterized by

*Figure 1 continued on next page*

*Figure 1 continued*

oocyte loss. Germ cells/oocytes are shown in blue. Dashed line: medullary–cortical boundary. (**B**) Postnatal follicular developmental timeline: wave 1 antral follicles emerge by 3 wk (*Figure 1—figure supplement 1A–C*); juvenile sexual development (peri-puberty) occurs at 4–5 wk; ovulation takes place around 7–8 wk. (**C**) Annotation of total follicles in P5 and P10 ovaries using Imaris software (gray spheres). Each follicle with a DDX4-positive oocyte was manually counted and labeled using the Imaris spot model. Scale bars: left, 50 µm; right, 70 µm. (**D**) Total follicle counts in P5 and P10 ovaries (mean ± SD; N = 3–4). ns, not significant. (**E**) Annotation of wave 1 follicles in P5 and P10 ovaries using Imaris software (gray spheres). Follicles with oocytes larger than 20 µm in diameter, primarily located in the medulla, are classified as wave 1 follicles at the indicated times. Each gray sphere represents one follicle. Scale bars: left, 50 µm; right, 70 µm. (**F**) Wave 1 follicle counts in P5 and P10 ovaries (mean ± SD; N = 3–4). ns, not significant. (**G**) 3D projection of the ovary at P14, reconstructed using the Imaris surface model. Scale bars = 50 µm. (**H**) Oocyte volume increase in wave 1 follicles over time (mean ± SD; N = 10). (Note: Nuclear volume is excluded, as DDX4 is a cytoplasmic protein.) ****p < 0.0001. (**I**) Representative images of ovaries at 4 and 5 wk. Left (4 wk): arrowheads indicate an atretic follicle with a distorted oocyte and disorganized granulosa cells; arrow indicates a developing healthy secondary follicle. Right (5 wk): arrowheads indicate degrading follicles at various stages; arrow indicates a developing healthy secondary follicle. Scale bars = 100 µm. (**J**) Wave 1 follicle remodeling, indicated by oocyte loss over time (mean ± SD; N = 3–4). Oocyte loss begins around 3 wk and continues through 5 wk, accompanied by granulosa cell disorganization and cell death. ns, not significant; **p < 0.01; ****p < 0.0001.

The online version of this article includes the following figure supplement(s) for figure 1:

**Figure supplement 1.** Wave 1 follicle development in the early juvenile ovary.

**Figure supplement 2.** Wave 1 follicles begin to remodel by 3 wk.

genes when granulosa cells are still reduced in number by atresia. The granulosa cell gene expression in developing wave 1.5 follicles also shows differences from that of wave 1 follicles, indicating that the distinct developmental fates of wave 1 follicles and of follicles that develop after a period of arrest (wave 1.5 and 2 follicles) may depend on intrinsic gene expression differences in addition to differing levels of gonadotropins.

## Results

### Wave 1 follicle development

Advances that allow tissue clearing (*Faire et al., 2015*; *Feng et al., 2017*; *Fiorentino et al., 2020*; *McKey et al., 2022*) now make it possible to image an entire mouse ovary without sectioning. To better understand follicle waves, we employed whole-mount staining combined with the clearing-enhanced 3D (C$_e$3D) method to analyze wild-type C57BL/6J ovaries at the cellular level (*Li et al., 2017*). Multiple ovaries were reconstructed during 1–5 wk of development (*Table 1*). In C57BL/6J mice, puberty typically occurs between 4 and 5 wk of age, but the first ovulation, marking sexual maturity, does not take place until 7 wk (*Figure 1B*; *Hogan et al., 1994*; *Nelson et al., 1990*; *Zhou et al., 2007*).

These studies allowed the number of wave 1 and 2 follicles to be accurately determined. Wave 1 follicles begin to grow into primary follicles around the time of birth (*Figure 1—figure supplement 1A, B*; *Videos 1 and 2*), as confirmed by gene expression analysis (*Niu and Spradling, 2020*). Following whole-ovary imaging using a confocal microscope, the samples were analyzed with Imaris

**Table 1.** Summary of whole-mount analyses of C57BL/6J ovaries.

| Postnatal age | Total follicles/ovary | Antral follicles/ovary | Wave 1 follicles/ovary | Reference files | Ovaries analyzed |
|---|---|---|---|---|---|
| P5–P7 | 3822 ± 838 | 0 | 278 ± 17 | *Video 1*.mp4 *Video 2*.mp4 | 4 |
| P10–P14 | 3304 ± 926 | 0 | 293 ± 36 | *Video 3*.mp4 *Video 4*.mp4 | 3 |
| 3 wk | 3089 ± 669 | 23 ± 2 | ND | *Video 5*.mp4 *Video 6*.mp4 | 3 |
| 4 wk | ND | 35 ± 2 | ND | *Video 7*.mp4 *Video 8*.mp4 | 4 |
| 5 wk | ND | 24 ± 4 | ND | *Video 9*.mp4 *Video 10*.mp4 | 5 |

Values are presented as mean ± SD. Data were collected from at least two litters, and the experiment was repeated once. ND, not determined.

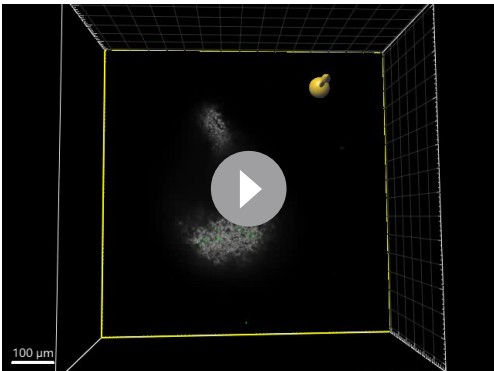

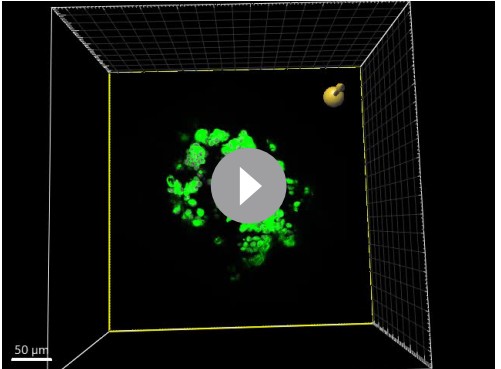

**Video 1.** Whole ovary from a 1-wk mouse. The ovary was whole-mount stained, cleared, and imaged using a Leica SP8 inverted microscope with a 20× objective. The 3D projection was generated in Imaris. Gray spheres indicate wave 1 follicles annotated using the Imaris spot model. DDX4: green; DAPI: white. Scale bar = 100 μm.

https://elifesciences.org/articles/107352/figures#video1

**Video 2.** Partial ovary from a 1-wk mouse. A field of the ovary was imaged with a Leica SP8 inverted microscope using a 40× objective after whole-mount staining and clearing. The Imaris surface model was used to reconstruct oocyte contours. DDX4: green; DAPI: white. Scale bar = 50 μm.

https://elifesciences.org/articles/107352/figures#video2

software. Since nearly all oocytes are enclosed by granulosa cells by P5, it becomes feasible to accurately quantify the number of follicles. At P5 and P10, we counted approximately 3822 and 3304 total follicles per ovary, respectively (*Figure 1C, D*), and 278 and 293 wave 1 follicles (*Figure 1E, F*). Despite variations in total follicle numbers among samples of the same age, the number of wave 1 follicles remained relatively consistent.

By 2 wk, wave 1 follicles in the medulla had developed into secondary follicles containing two to four layers of granulosa cells and one to two layers of theca cells (*Figure 1—figure supplement 1C*; *Videos 3 and 4*). At 3 wk, some follicles further developed into antral follicles with multiple granulosa cell layers, within which granulosa cell apoptosis was observed (*Figure 1—figure supplement 1D*). Using Imaris surface modeling, we quantified oocyte volume from 1 to 3 wk and observed a significant increase during this period (*Figure 1G, H*).

## Wave 1 follicles undergo partial atresia with theca cell survival

Starting around 3 wk, rather than continuing to develop, wave 1 follicles begin to undergo atresia (*Table 1*; *Videos 5–10*). We investigated the kinetics, cell specificity, and completeness of cellular

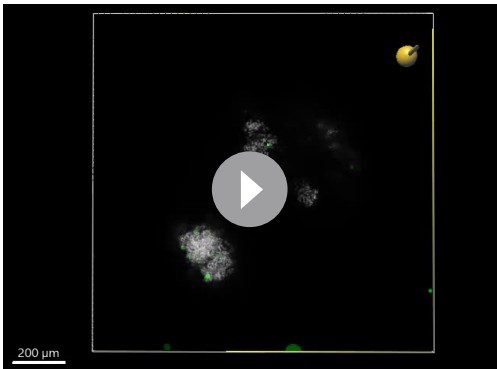

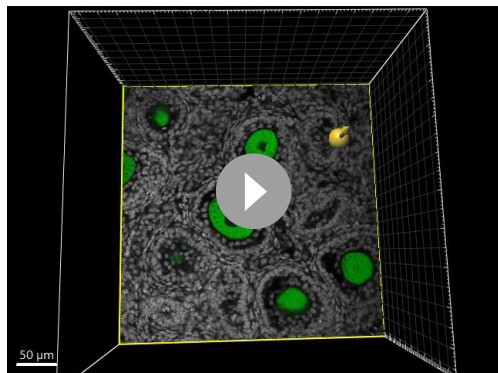

**Video 3.** Whole ovary from a 2-wk mouse. The ovary was whole-mount stained, cleared, imaged with a Leica SP8 inverted microscope, and projected in Imaris. Gray spheres indicate wave 1 follicles using the Imaris spot model. DDX4: green; DAPI: white. Scale bar = 200 μm.

https://elifesciences.org/articles/107352/figures#video3

**Video 4.** Partial ovary from a 2-wk mouse. The ovary was imaged with a Leica SP8 inverted microscope using a 40× objective after whole-mount staining and clearing. The Imaris surface model shows oocyte contours. DDX4: green; DAPI: white. Scale bar = 50 μm.

https://elifesciences.org/articles/107352/figures#video4

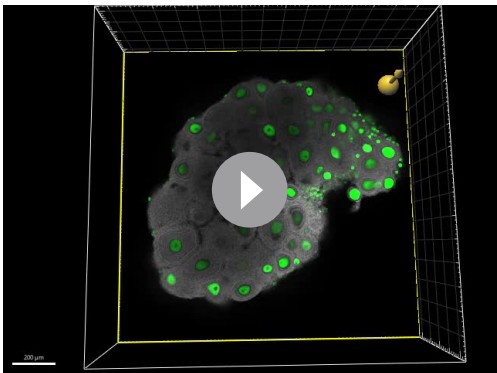

**Video 5.** Partial ovary from a 3-wk mouse. The ovary was whole-mount stained, cleared, imaged with a Leica SP8 inverted microscope using a 10× air objective, and projected in Imaris. DDX4: green; DAPI: white. Scale bar = 200 μm.

https://elifesciences.org/articles/107352/figures#video5

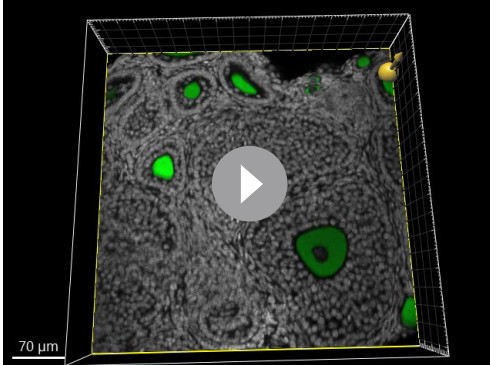

**Video 6.** Partial ovary from a 3-wk mouse. A field of the ovary was imaged using a Leica SP8 inverted microscope with a 40× objective after whole-mount staining and clearing. The Imaris surface model shows oocyte contours. DDX4: green; DAPI: white. Scale bar = 70 μm.

https://elifesciences.org/articles/107352/figures#video6

turnover from 3 to 5 wk. At 4 wk, atretic follicles initially showed oocyte distortion followed by loss, leaving behind disorganized granulosa cell layers within the basement membrane but without affecting the theca cell layers (*Figure 1I*, left). By 5 wk, atretic follicles progressively decreased in size and developed a cavity beneath the granulosa cell layer where the oocyte had previously resided (*Figure 1I*, right). Oocyte loss from wave 1 follicles increased significantly between 3 and 4 wk, exceeding 80% by 5 wk (*Figure 1J*). Electron microscopy revealed that during atresia, as oocytes degenerate, microvilli from both oocytes and granulosa cells retract from the zona pellucida (ZP). The shrinking of the cavity is marked by folding of the ZP, and in atretic follicles, remnants of the ZP appear to prevent further collapse of the cavity structure (*Figure 1—figure supplement 2A*). ZP remnants in the follicular cavity have previously been used as a morphological indicator of atretic follicles (*Myers et al., 2004*). We confirmed that these events also occur in the DBA/2J strain, where atretic follicles are first observed around 3 wk (*Figure 1—figure supplement 2B*, left). By 7 wk, corpora lutea from ovulated oocytes and cavities left behind by wave 1 oocyte loss were evident (*Figure 1—figure supplement 2B*, right). Thus, by combining whole-mount staining with the C$_e$3D clearing method, we found that the great majority of wave 1 follicles lose their

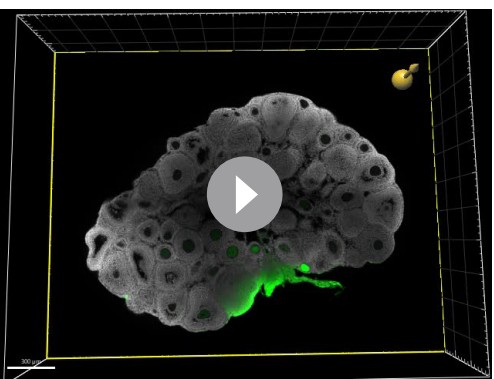

**Video 7.** Partial ovary from a 4-wk mouse. The ovary was whole-mount stained, cleared, imaged with a Leica SP8 inverted microscope using a 20× immersion objective, and projected in Imaris. DDX4: green; DAPI: white. Scale bar = 300 μm.

https://elifesciences.org/articles/107352/figures#video7

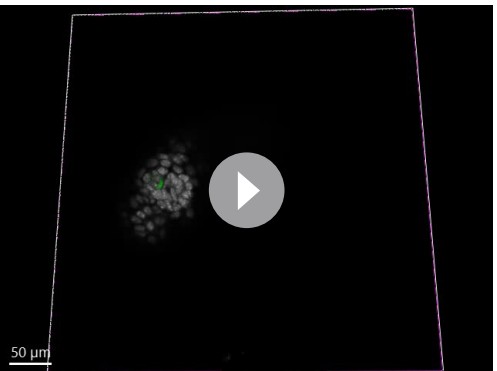

**Video 8.** Partial ovary from a 4-wk mouse. A field of the ovary was imaged with a Leica SP8 inverted microscope using a 40× objective after whole-mount staining and clearing. The Imaris surface model shows oocyte contours. DDX4: green; DAPI: white. Scale bar = 50 μm.

https://elifesciences.org/articles/107352/figures#video8

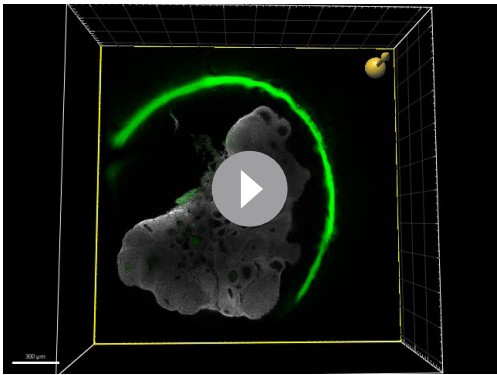

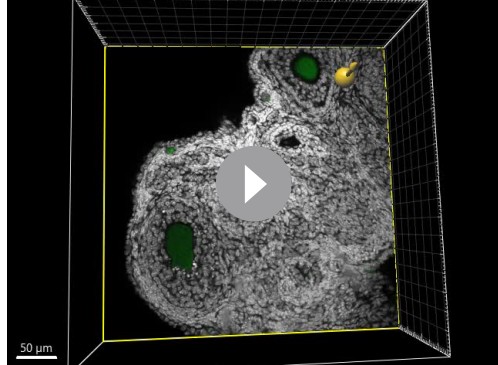

**Video 9.** Partial ovary from a 5-wk mouse. The ovary was whole-mount stained, cleared, imaged with a Leica SP8 inverted microscope using a 20× immersion objective, and projected in Imaris. DDX4: green; DAPI: white. Scale bar = 300 µm.
https://elifesciences.org/articles/107352/figures#video9

**Video 10.** Partial ovary from a 5-wk mouse. A field of the ovary was imaged using a Leica SP8 inverted microscope with a 40× objective after whole-mount staining and clearing. DDX4: green; DAPI: white. Scale bar = 50 µm.
https://elifesciences.org/articles/107352/figures#video10

oocytes prior to puberty and do not contribute to early ovulation.

The loss of oocytes from most wave 1 follicles is not simply due to atresia of follicles whose oocytes are defective, as superovulation at these stages can rescue many of them (*Lamas et al., 2021*). Etymologically, atresia originates from the Greek 'a-' (meaning 'no') and 'tresis' (meaning 'perforation'), referring to the absence or closure of an opening or cavity. The term is sometimes used simply to describe the complete turnover of follicles, which was not observed in wave 1 follicles. We prefer the terms 'partial atresia' or 'remodeled follicles' to emphasize the potential function of the structures that remain (*Hsueh et al., 1994*).

To analyze the cell specificity of wave 1 partial atresia, we used the *Foxl2-CreERT2* mouse line crossed with the *Rosa26-LSL-EYFP* reporter line to selectively lineage-label granulosa cells associated with wave 1 follicles (*Figure 2A*). Similar genetic models have previously been used to study wave 1 follicle development and the duration of their fertility contribution based on tissue sections (*Mork et al., 2012*; *Zheng et al., 2014a*). Following brief tamoxifen (TAM) administration at embryonic day 16.5 (E16.5), we observed a distinct labeling pattern along the medullary–cortical boundary. Most wave 1 follicles in the medulla showed mosaic EYFP labeling of granulosa cells at 2 wk (due to the incomplete efficiency of Cre-Lox recombination), while granulosa cells in the largely cortical wave 2 follicles remained unlabeled (*Figure 2B, C*). On average, 201 preantral wave 1 derivatives—representing approximately 90% of wave 1 follicles—exhibited mosaic labeling in granulosa cells (*Figure 2E, F*). By 5 wk, after extensive wave 1 atresia and remodeling, an average of only 26 antral follicles containing healthy oocytes and EYFP-mosaic granulosa cell layers were observed, accounting for approximately 70% of all antral follicles (*Figure 2D–F*).

By comparing the number of labeled granulosa cells in healthy follicles at 3 wk with those in remodeled follicles at 5 wk (*Figure 2G*, left), we found that remodeled follicles had lost not only their oocytes but also 94% of their granulosa cells (*Figure 2G*, right). The remaining granulosa cells were located adjacent to the cavity left by the oocyte, and approximately 80% of remodeled follicles still contained labeled granulosa cells (*Figure 2H*). To test whether these remnant granulosa cells remained active, lineage labeling was performed at 5 wk and analyzed one week later. The results showed that many granulosa cells in remodeled follicles persisted adjacent to cavities left by oocyte loss and continued to express *Foxl2* (*Figure 2I*). The turnover of wave 1 follicles was also confirmed using a different reporter mouse line, *Rosa26-LSL-tdTomato* (*Figure 2—figure supplement 1A–C*).

## Identification of an early activating subpopulation of primordial follicles

Because only about 26 normal-appearing wave 1 follicles with oocytes survived to 5 wk, which seemed too few to support the months of early fertility ascribed to wave 1 follicles by a previous report (*Zheng et al., 2014a*), we looked for follicles other than wave 1 that might also be labeled by *Foxl2* expression.

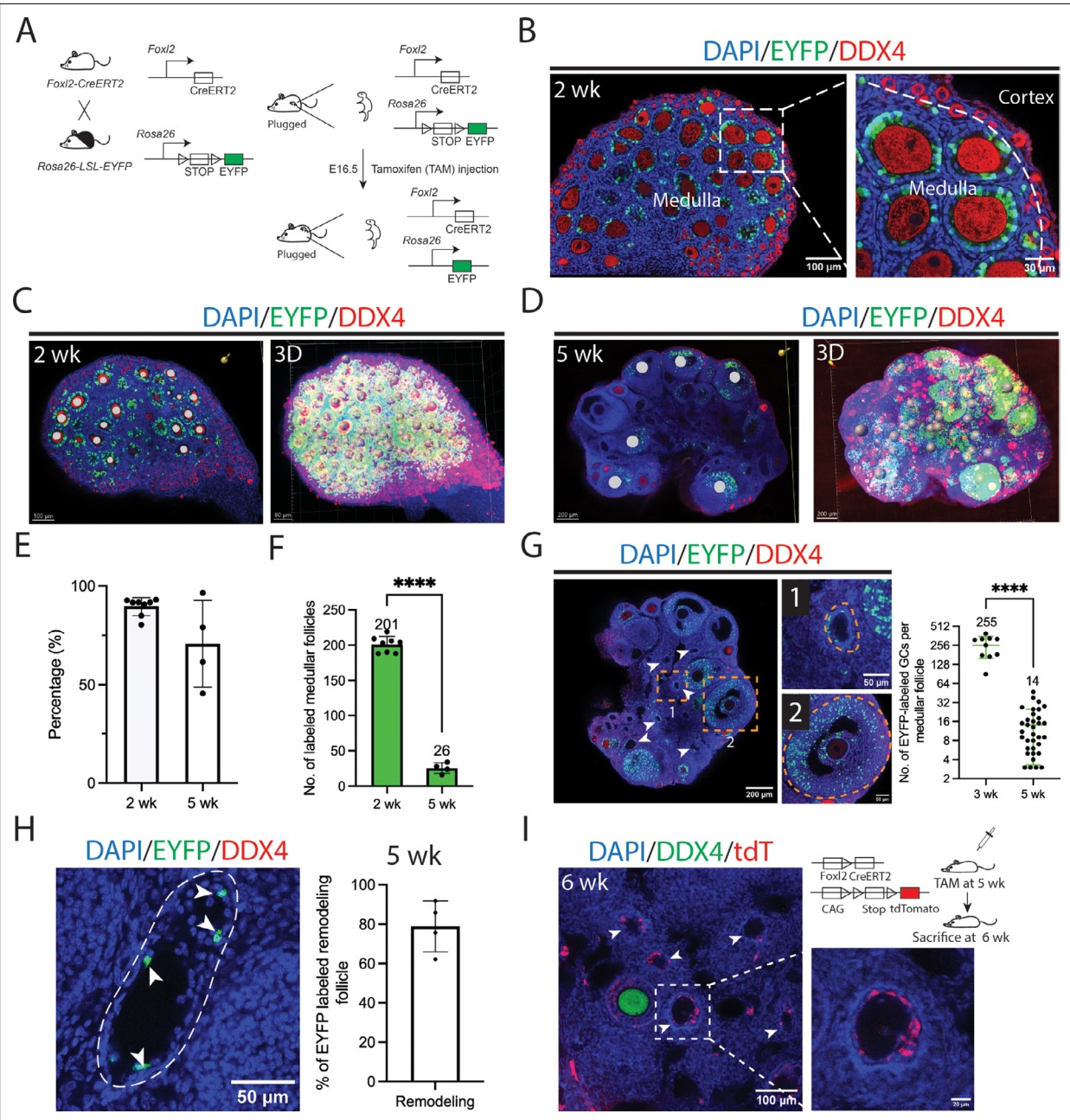

**Figure 2.** Wave 1 follicles lose granulosa cells during remodeling. (**A**) Schematic diagram illustrating activation of *Foxl2*-driven EYFP expression following TAM injection at E16.5. TAM administration at E16.5 induces excision of a stop cassette, leading to EYFP expression in granulosa cells of follicles (see Methods). (**B**) Efficient EYFP labeling of granulosa cells in wave 1 medullary follicles of a 2-week ovary following TAM injection at E16.5, as shown in (**A**), magnified on the right. Scale bars: left, 100 µm; right, 30 µm. (**C**) EYFP-labeled follicles that reached the primary/secondary follicle stage by 2 wk were quantified using Imaris software, with annotation performed using the spot model. Left: section showing annotated labeled follicles (solid white circles). Right: 3D projection using Imaris software, with labeled follicles represented by gray spheres. Scale bars: left, 100 µm; right, 80 µm. (**D**) By 5 wk, EYFP-labeled follicles that reached the preantral/antral follicle stage were quantified using Imaris software and annotated using the spot model. Left: section showing annotated labeled follicles (solid white circles). Right: 3D projection using Imaris software, with labeled follicles represented by gray spheres. Scale bars = 200 µm. (**E**) Over 90% of wave 1 follicles contained EYFP-positive granulosa cells at 2 wk. At 5 wk, approximately 70% of normal antral follicles contained EYFP-positive granulosa cells. (**F**) Quantification of wave 1 follicle numbers (representative images shown in C–D) shows a decline from 2 to 5 wk. ****p < 0.0001. Each dot represents one sample. (**G**) Representative section from a 5-week ovary showing ongoing remodeling of wave 1 follicles. Follicles near the medullary core appear small and without oocytes (arrowheads), with few remaining EYFP-labeled granulosa cells, as seen in the follicle in rectangle '1', enlarged on the right. In contrast, a normal antral follicle in rectangle '2', also enlarged on the right, contains numerous EYFP-labeled granulosa cells. Right: average number of EYFP-labeled granulosa cells in medullary follicles before remodeling (3 wk) versus

*Figure 2 continued on next page*

*Figure 2 continued*

remodeled medullary follicles at 5 wk. Each point represents one analyzed follicle. ****p < 0.0001. Scale bars: left, 200 μm; right, 50 μm. (**H**) About 80% of remodeled follicles retained some EYFP-labeled granulosa cells at 5 wk, consistent with their wave 1 origin. Dashed line: follicle boundary; arrowheads: EYFP-positive granulosa cells. Each point represents one analyzed ovary. (**I**) Remnant granulosa cells of remodeled wave 1 follicles continued to express *Foxl2* at 5 wk. TAM was injected into *Foxl2-CreERT2; Rosa26-LSL-tdTomato* mice at 5 wk, and ovaries were collected 1 week later. Arrowheads indicate remodeled follicles; one follicle (outlined in the rectangle) is enlarged on the right. Scale bars: left, 100 μm; right, 20 μm.

The online version of this article includes the following figure supplement(s) for figure 2:

**Figure supplement 1.** Lineage labeling using *Foxl2-CreERT2* and *Rosa26-LSL-tdTomato*.

We observed that *Foxl2-CreERT2* activated by TAM injection at E16.5 in 2 wk juvenile ovaries also labeled a subset of primordial follicles located near the medullary–cortical boundary (*Figure 3A*). A mean of 421 labeled 'boundary' follicles, each mosaic for 1–4 EYFP-positive granulosa cells, was identified among 5843 total follicles (7%) (*Figure 3B*). Because only one to four granulosa cells were labeled from among an average of eight granulosa cells (*Chen et al., 2022*), additional boundary follicles were likely missed for stochastic reasons due to the low efficiency of Cre-Lox recombination. Applying the Poisson distribution based on the relative proportion of follicles with one or two labeled granulosa cells suggested that 90% of boundary follicles were detected.

Because of the biological interest of this subgroup of primordial follicles, we repeated activation of *Foxl2-CreERT2* at E16.5 in a separate experiment using a different reporter line, *Rosa26-LSL-tdTomato*. The results confirmed the existence and properties of boundary follicles (*Figure 2—figure supplement 1D*). Boundary follicles were also identified following TAM injection at E14.5, although at a lower frequency (*Figure 3C*). Thus, the presence of this subclass was validated in multiple experiments.

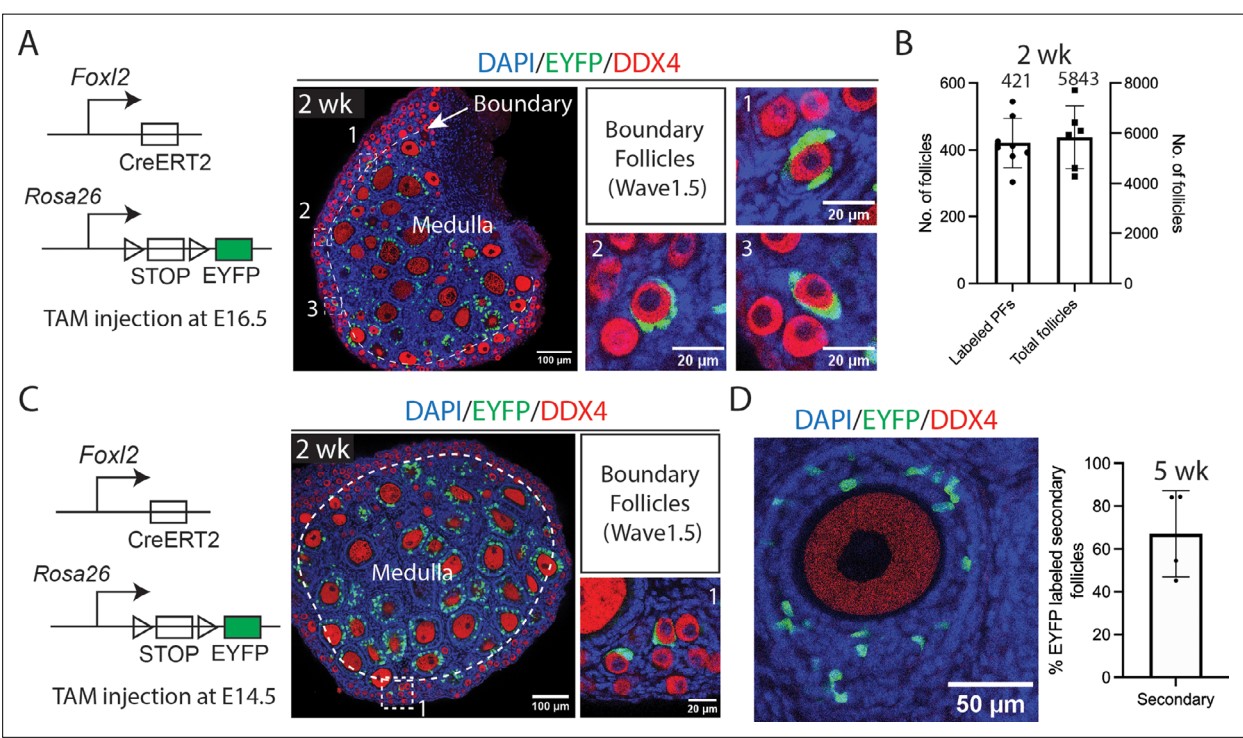

**Figure 3.** The labeling and quantification of boundary follicles. (**A**) At 2 wk, one or more granulosa cells in some primordial follicles located at the medullary–cortical boundary are labeled with EYFP (examples enlarged in '1', '2', '3') following TAM injection, as shown in the schematic. These follicles are referred to as boundary follicles or, in accordance with wave classification, wave 1.5 follicles. Dashed line: medullary–cortical boundary. Scale bars: left, 100 μm; right, 20 μm. (**B**) Quantification shows that 7% (421 ± 74) of the total 5843 ± 1256 primordial follicles contained EYFP-labeled granulosa cells in the experiment shown in (**A**). Each point represents one analyzed sample. Labeled PFs: labeled primordial follicles. (**C**) Representative images of *Foxl2-CreERT2; Rosa26-LSL-EYFP* mouse ovary injected with TAM at E14.5. Boundary follicles labeled with EYFP are enlarged on the right. Dashed line: medullary–cortical boundary. Scale bars: left, 100 μm; right, 20 μm. (**D**) Approximately 70% of secondary follicles in 5-wk ovaries contained EYFP-labeled granulosa cells, suggesting they originate from labeled primordial follicles (wave 1.5) at 2 wk. TAM was injected at E16.5. Each point represents one analyzed sample. Scale bar = 50 μm.

Primordial follicles begin to activate after 2 wk (*Zheng et al., 2014a*). If boundary follicles behaved like other primordial follicles, we would expect that only 7% (421/5823) of newly activated wave 2 growing follicles would show EYFP-mosaic granulosa cells at later times. Instead, we found that among secondary follicles at 5 wk, nearly 70% contained mosaic EYFP-positive granulosa cells (*Figure 3D*). Thus, boundary follicles activate at the time of the earliest known primordial follicles and rise from 7% of quiescent to >70% of early-activating follicles, implying that they comprise most of the earliest activated primordial follicles. Moreover, they correspond in location and properties to the inactive but poised primordial follicles described by *Meinsohn et al., 2021a*; *Meinsohn et al., 2021b*.

## Partial atresia of wave 1 follicles does not affect theca cells, which expand in number

We then used lineage labeling to investigate the fate of wave 1 theca cells after the onset of atresia. *Cyp17a1* is an essential gene involved in steroidogenesis and is expressed in the steroidogenic theca cells and interstitial gland cells. We validated that these cells strongly expressed the steroid biosynthetic enzyme CYP17A1 from 3 to 5 wk using immunofluorescence staining (*Figure 4—figure supplement 1A*). To lineage-label these cells, we crossed the *Cyp17a1-iCre* transgenic mouse line with the *Rosa26-LSL-tdTomato* reporter line (*Figure 4A*). In 2-wk ovaries, tdTomato expression was observed in theca cells sheathing large medullary follicles (*Figure 4B*; arrowhead). Additional cells intertwined with theca cells, which appeared to be interstitial gland cells (*Figure 4B*; boxed region), were visualized using Imaris and pseudo-colored using the surface model to highlight their structure (*Figure 4C*). A relatively small number of granulosa cells were also initially labeled (*Figure 4—figure supplement 1B*, arrow), possibly because granulosa cell precursors transiently express *Cyp17a1* during embryonic stages (*Mawaribuchi et al., 2014*).

Between 2 and 5 wk, the number of tdTomato-labeled cells associated with follicular cavities increased dramatically as wave 1 follicle atresia and remodeling proceeded to generate more such follicles (*Figure 4D, E*). Because over 90% of granulosa cells turn over as part of wave 1 remodeling (*Figure 2G*), the tdTomato-positive cells associated with remodeled wave 1 follicles by 5 wk mostly represented abundant steroidogenic theca cells and interstitial gland cells. Oil Red O staining revealed abundant lipid droplet accumulation in these cells from 2 to 5 wk (*Figure 4F, G*). Antibody staining confirmed abundant expression of the steroidogenic enzyme HSD3B1 and the lipid storage protein PLIN1 in these cells, which were located in the medulla (*Figure 4H*). Electron microscopy further demonstrated that expanded wave 1 follicular cells display distinctive lipid droplets and mitochondria characteristic of steroidogenic theca and interstitial gland cells (*Figure 4I*, *Figure 4—figure supplement 1C*). Using the Imaris software surface model, we reconstructed labeled steroidogenic theca and interstitial gland cells throughout the entire ovary. By 5 wk, atretic wave 1 follicles had generated a cellular cluster within the medulla dominated by expanded numbers of steroidogenic cells (*Figure 4J*). This structure persisted in the ovary at least until 7 wk (*Figure 4—figure supplement 1D*).

## Mapping cell transcription during wave 1 remodeling

To gain further insight into wave 1 follicle development and remodeling, we analyzed single-cell gene expression from ovaries collected weekly between 2 and 6 wk, covering pre- and peri-pubertal stages (*Figure 5A*; *Source data 1–6*). A total of 31 initial clusters (c0–c30) were identified, with nearly all samples contributing to each cluster (*Figure 5B*). Nine major cell groups were identified (*Figure 5C*) based on marker gene expression (*Figure 5D*; *Niu and Spradling, 2020*; *Morris et al., 2022*). These included 10 granulosa cell clusters and 8 stromal cell clusters, as well as theca cells, hematopoietic cells, endothelial cells, pericytes, smooth muscle cells, and epithelial cells. Four clusters with low UMIs (c6, c11, c15, and c29) were excluded from subsequent analysis (*Figure 6—figure supplement 1A*). Additionally, cluster c14 was removed due to potential doublets, as it overlapped with several other clusters; however, we cautiously named it 'TBD (to be determined)" (*Figure 6—figure supplement 1B*). Notably, all known ovarian somatic cell types were identified within one or more clusters. Germ cells, however, were likely too rare in these juvenile ovaries to form a cluster, although ovarian germ cells have been extensively studied at earlier developmental stages by these methods (*Niu and Spradling, 2022*) or at similar stages with alternative approaches (*Morris et al., 2022*; *Isola et al., 2024*).

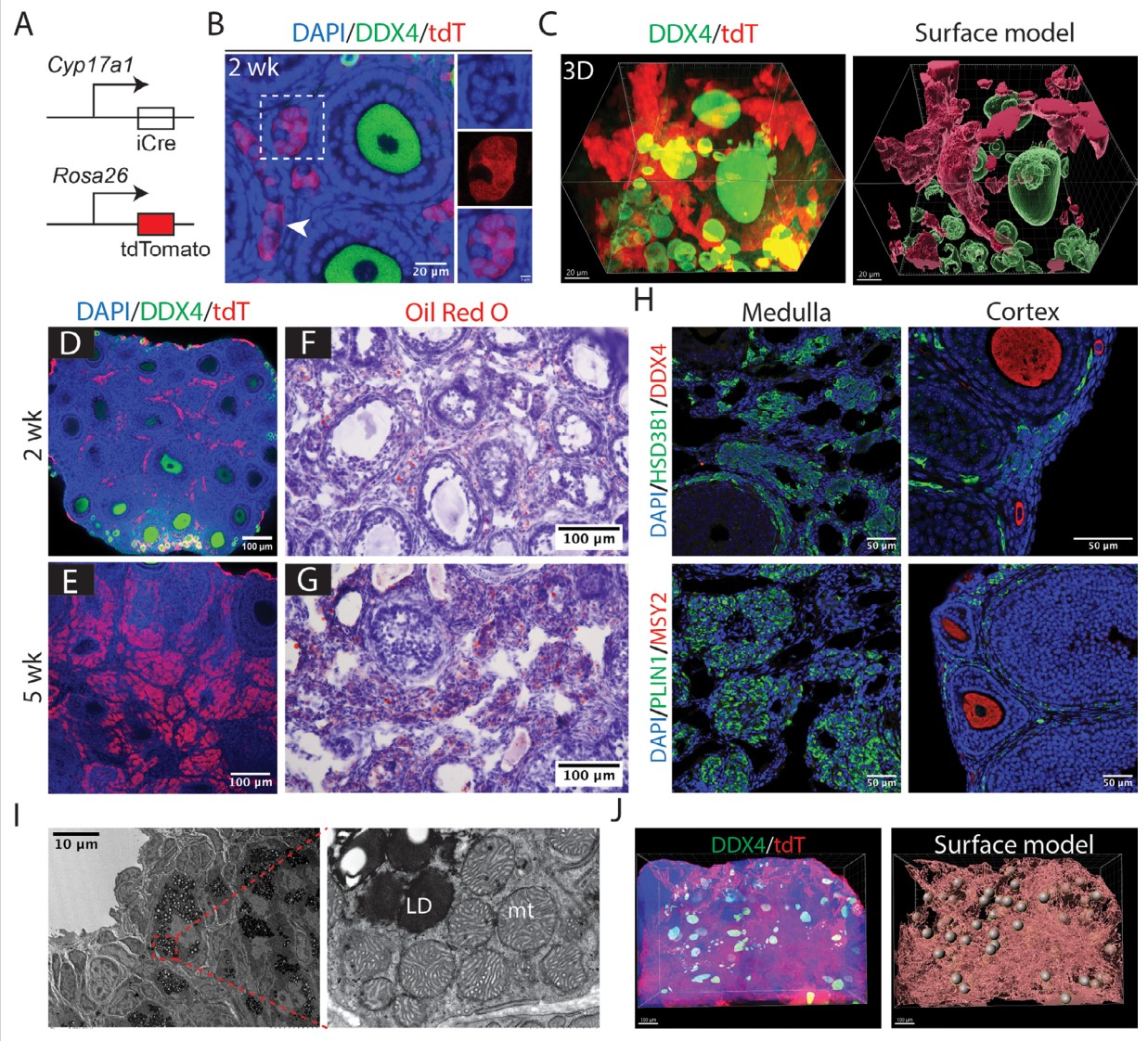

**Figure 4.** Follicles remodel to form an interconnected stromal network rich in thecal secretory cells. (**A**) Schematic diagram illustrating lineage labeling of *Cyp17a1*-expressing theca cells, interstitial gland cells, and potentially other cell types using the *Cyp17a1*-iCre; *Rosa26-LSL-tdTomato* mouse model. (**B**) A 2-wk ovary showing tdTomato-labeled theca cells sheathing follicles (arrowhead) and putative interstitial gland cells between follicles (rectangle). Scale bars: left, 20 µm; right, 5 µm. (**C**) 3D reconstruction of follicles with surrounding labeled theca cells and interconnected interstitial gland cells in a 2-wk ovary. Left: 3D projection using Imaris software. Right: pseudo-colored annotation using the Imaris surface model. Green: oocytes. Red: *Cyp17a1*-expressing cells, including theca and interstitial gland cells. Scale bars = 20 µm. Representative images of 2 wk (**D**) and 5 wk (**E**) ovaries showing a dramatic increase in labeled theca cells, interstitial gland cells, and possibly other *Cyp17a1*-lineage cells in the ovarian medulla. Scale bars = 100 µm. Oil Red O staining (red) reveals a significant increase in lipid droplet storage in ovarian medullary cells from 2 wk (**F**) to 5 wk (**G**) ovaries. Scale bars = 100 µm. (**H**) Immunofluorescence staining of HSD3B1 and PLIN1 at 5 wk shows that genes required for steroidogenesis and lipid droplet storage are highly expressed in medullary cells. Scale bars = 50 µm. (**I**) Electron microscopy of a remodeled follicle reveals cells with abundant lipid droplets (LD) and characteristic mitochondria (mt) (boxed region magnified in right panel). Scale bar = 10 µm. (**J**) 3D reconstruction of a 5-wk ovary labeled by tdTomato expression activated by *Cyp17a1-iCre*. Left: 3D projection using Imaris software. Right: pseudo-colored using the Imaris surface model. Red: theca/interstitial network structure. Spheres: oocyte locations. Scale bars = 100 µm.

The online version of this article includes the following figure supplement(s) for figure 4:

**Figure supplement 1.** Expansion of steroidogenic theca cells and interstitial gland cells.

The initial clustering further validated the picture of wave 1 development obtained from lineage studies. Cluster c1 expressed wave 1 BPG markers such as *Nr5a2*, *Hsd3b1*, *Inha*, and *Inhba*, and its cell percentage increased approximately 3.5-fold between 2 and 4 wk before falling at 5 wk as wave 1 granulosa cells turned over (*Source data 6*). Cluster c16 highly expressed theca cell steroidogenic

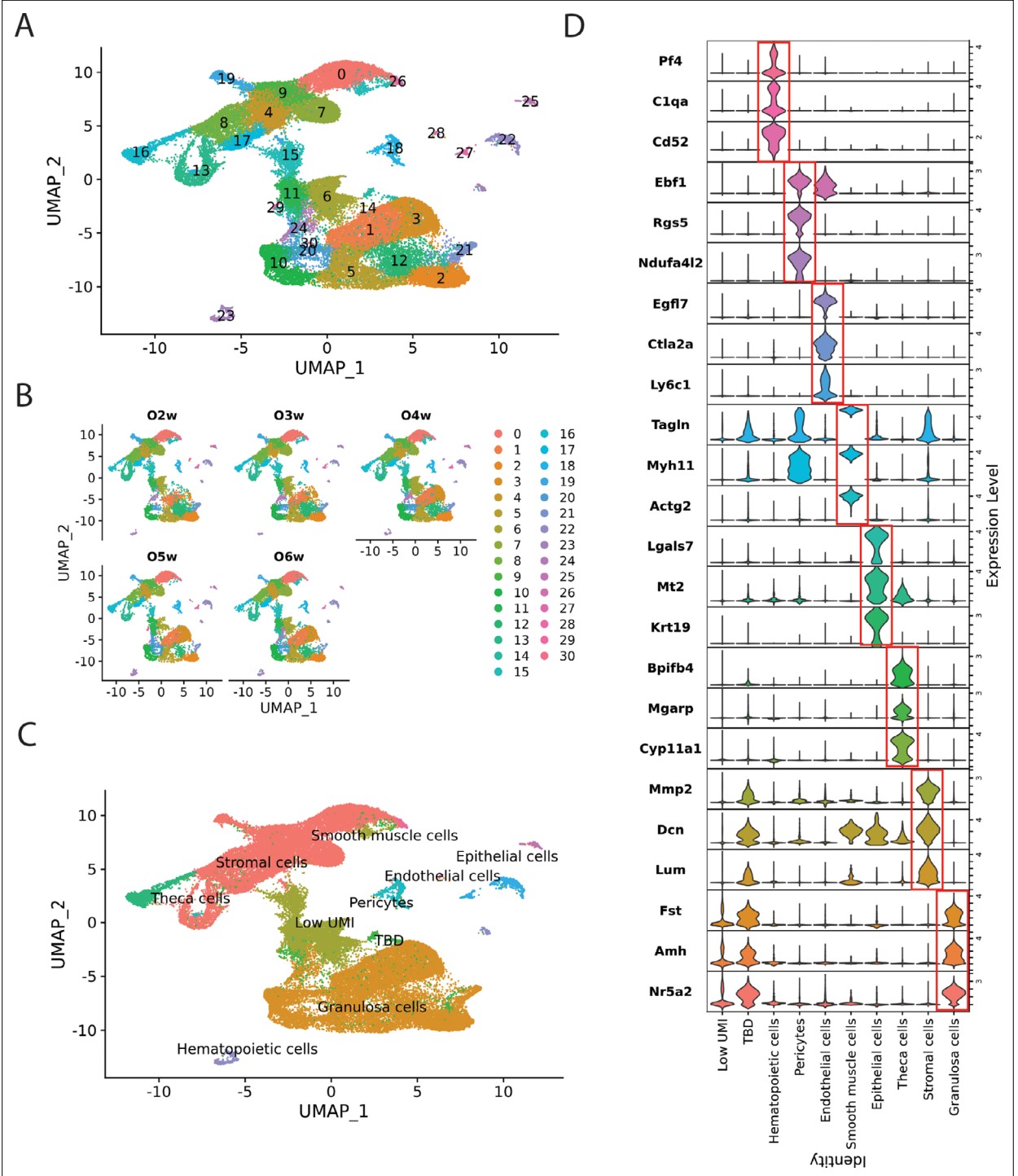

**Figure 5.** Single-cell RNA sequencing of 2- to 6-wk ovaries. (**A**) UMAP plot of ovaries from 2 to 6 wk, showing 31 initial clusters (c0–c30) represented in different colors. Sequencing data from different ages were merged before being analyzed in Seurat (see Methods for details; see *Source data 1*). (**B**) Ovarian cells from 2 to 6 wk contribute to almost every cluster in the UMAP plot. O2w: scRNA-seq data from 2-wk ovary. O3w: 3-wk ovary. O4w: 4-wk ovary. O5w: 5-wk ovary. O6w: 6-wk ovary. (**C**) Cell type groups are indicated by colored regions, with their relationship to the numbered initial clusters as shown in (**A**). Granulosa cells: c1, c2, c3, c5, c10, c12, c20, c21, c24, and c30; stromal cells: c0, c4, c7, c8, c9, c13, c17, and c19; theca cells: c16; hematopoietic cells: c23; endothelial cells: c22 and c27; pericytes: c18; smooth muscle cells: c26; epithelial cells: c25 and c28; low UMI clusters: c6, c11, c15, and c29; TBD (to be determined): c14. (**D**) Multi-violin plot of selected marker gene expression across different cell types. *Y*-axis: gene names with expression levels normalized for display. *X*-axis: cell types.

genes such as *Cyp11a1*, *Cyp17a1*, and *Plin1*. The number of c16 cells peaked at puberty (5 wk). The outer 'theca externa' is enriched in angiogenic cells, blood vessels, and fibroblastic cells. Clusters corresponding to endothelial cells expressing *Flt1* (c22) and blood cells expressing *Laptm5* (c23) increased nearly fourfold between 3 and 5 wk, consistent with such vascularization.

## Analyzing granulosa cell gene expression

We re-clustered the 10 initial granulosa cell clusters in Seurat, yielding 18 new clusters (gc0–gc17) (*Figure 6A*). These were classified into groups corresponding to their likely association with primordial (gc11), primary (gc3 and gc8), secondary (gc1, gc4, and gc9), antral (gc2, gc7, gc10, and gc13), and remodeling (gc0 and gc6) follicles, as well as mitotically cycling cells (gc5, gc12, gc14, gc15, gc16, and gc17) (*Figure 6B, C*). Granulosa cells from all follicle stages were identified across all samples (*Figure 6—figure supplement 1C*). Single-cell pseudotime trajectory analysis using Monocle 3 identified distinct developmental paths from primordial follicles to either antral follicles or remodeled follicles (*Figure 6D*). Once primordial follicles reached the secondary stage, some progressed along path 1 toward antral follicles, while others followed path 2 toward remodeled follicles.

We identified up to 250 top differentially expressed genes for each group (*Source data 5*) and performed further analysis using Metascape (*Zhou et al., 2019*). The development and remodeling of wave 1 follicles and of boundary follicles, giving rise to early wave 2 follicles, are summarized in *Figure 7*.

## Comparison of gene expression between wave 1 and 2 follicle development

Wave 1 and 2 follicles begin to differ in gene expression early in fetal development, as BPGs and EPGs express different genes as early as E14.5 and diverge further by P5 (*Niu and Spradling, 2020*). To investigate whether gene expression differences at follicle formation persist, and to look for additional wave-specific gene expression, we examined granulosa cell clusters gc2, gc7, gc10, and gc13 at 2 wk, comprising all granulosa cell clusters scored as in antral follicles (*Figure 6A, C*). As illustrated in *Figure 7*, only wave 1 antral follicles are present in significant numbers at 2 and 3 wk, prior to widespread wave 1 remodeling. In contrast, because of extensive wave 1 granulosa cell turnover, by 5 wk nearly all of these same antral follicle clusters originated from developing boundary follicles. Consequently, we compared gene expression levels during these two time intervals for all four antral granulosa cell clusters.

The nuclear receptor *Nr5a2* is preferentially expressed 7.8-fold higher at P5 in BPGs within growing wave 1 follicles compared to EPGs in growth-arrested wave 2 follicles (*Niu and Spradling, 2020*). Similarly, *Nr5a2* expression is 11-fold higher in all 2-wk antral follicle cell clusters compared to primordial follicle (gc11) granulosa cells. This difference reflects the early activation of wave 1 follicles, as *Nr5a2* turns on in all wave 2 granulosa cells as they begin to grow (*Meinsohn et al., 2019*). We observed that *Nr5a2* expression increases in primary and secondary follicles and reaches similar levels in wave 1 and 2 antral follicles (ratio = 1.02 ± 0.23). Comparable expression ratios were likewise seen between wave 1 and 2 antral granulosa cells for other genes that were BPG/wave 1-preferential at P5, including *Dhh* (1.04 ± 0.41), *Hsd17b1* (0.61 ± 0.14), *Ptgis* (1.08 ± 0.26), *Prlr* (1.8 ± 0.80), *Fshr* (0.89 ± 0.21), and *Igf1* (0.86 ± 0.10).

However, some genes continued to be differentially expressed between wave 1 and 2 follicles at the same stage of development. Wave 1 antral granulosa cells at 2 wk expressed *Wnt4* at much higher levels (4.8 ± 1.0) compared to wave 2 antral granulosa cells at either 5 wk (0.17 ± 0.069) or 6 wk (0.14 ± 0.070). Other genes expressed more highly in wave 1 than in wave 2 antral granulosa cells included *Igfbp5* (27 ± 10), *Itga6* (6.28 ± 0.97), and *Inha* (1.52 ± 0.29). Genes expressed at lower levels in wave 1 than in wave 2 included *Cald1* (0.20 ± 0.089), *Ghr* (0.16 ± 0.027), *Grb14* (0.23 ± 0.092), and *Mmp11* (0.46 ± 0.23). Thus, gene expression is not entirely equivalent in wave 1 and 2 follicles at corresponding stages, as has often been assumed.

Clusters gc0 and gc6 showed indications that they derived from remodeled wave 1 follicles. For example, the number of gc0 and gc6 cells was low at 2 and 3 wk before substantial atresia but increased 9.8-fold at 4 wk (*Source data 6*). Preferentially expressed genes in gc0 and gc6 included *Cyp1b1*, *Atf3*, *Cxcl1*, *Bmp2*, *Apoe*, and *Sox4*. Metascape analysis identified vasculature development (–7.7; –4.0), cell cycle (–6.6; –9.2), and positive regulation of programmed cell death (–4.1; nd), where

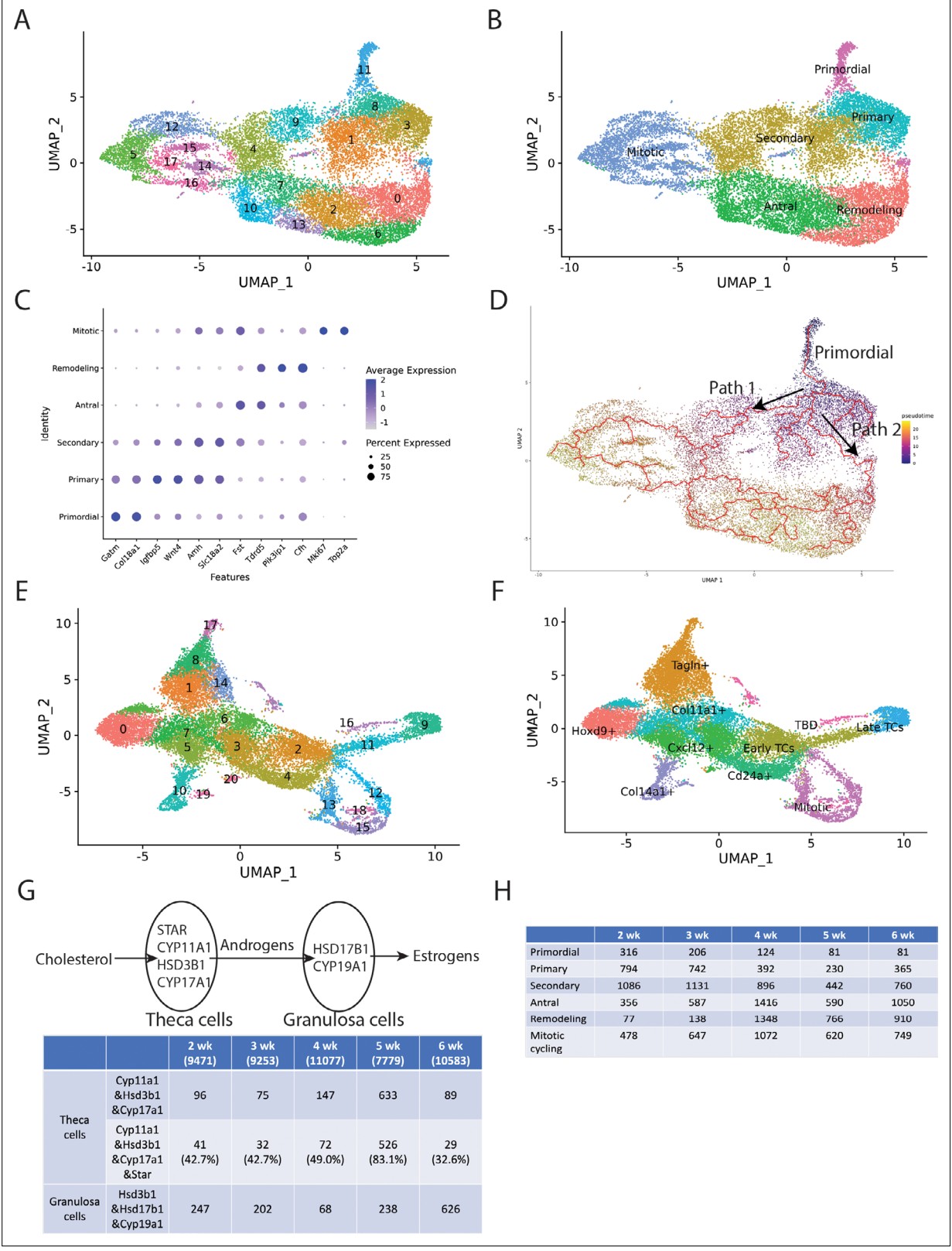

**Figure 6.** Further re-cluster analysis of sequencing data. (**A**) UMAP plot of re-clustered granulosa cells from *Figure 5A*, showing 18 clusters (gc0–gc17). (**B**) The deduced ovarian follicle developmental stages are labeled and indicated by colored regions, with their relationship to the numbered clusters shown in panel A. Primordial: gc11; primary: gc3 and gc8; secondary: gc1, gc4, and gc9; antral: gc2, gc7, gc10, and gc13; remodeling: gc0 and gc6; mitotic: gc5, gc12, gc14, gc15, gc16, and gc17. (**C**) Dot plot of marker gene expression in granulosa cells across different follicle stages. Y-axis: cell types.

*Figure 6 continued on next page*

*Figure 6 continued*

X-axis: gene names with expression levels normalized for display. (**D**) Pseudotime trajectory analysis using Monocle 3 reveals two pathways in secondary follicles: Path 1 leads to further development into antral follicles, while Path 2 results in remodeling follicles. (**E**) UMAP plot of re-clustered theca and stromal cells from *Figure 5A*, showing 21 clusters (g0–g20). (**F**) The deduced mesodermal cell subgroups are labeled and indicated by colored regions, with their relationship to the numbered clusters shown in panel E. Hoxd9+: g0; Tagln+: g1, g8, g14, and g17; early theca cells (TCs): g2 and g11; late theca cells (TCs): g9; Cxcl12+: g3, g5, and g20; Cd24a+: g4; Col11a1+: g6 and g7; Col14a1+: g10 and g19; mitotic: g12, g13, g15, and g18; TBD (to be determined): g16. (**G**) Diagram of steroidogenic gene expression in theca and granulosa cells and their role in androgen and estrogen synthesis. Top: Synthesis of androgens in theca cells and estrogens in granulosa cells from cholesterol. Bottom: Estimated numbers of steroidogenic theca cells and interstitial gland cells expressing three key steroidogenic genes in the sequencing data—*Cyp11a1*, *Hsd3b1*, and *Cyp17a1*—are shown, along with those expressing all four essential genes (*Cyp11a1*, *Hsd3b1*, *Cyp17a1*, and *Star*). The total cell numbers are shown in the first row. From 2 to 5 wk, the theca cell population increases, followed by a decline at 6 wk. Meanwhile, the number and percentage of theca cells expressing all four essential genes progressively increase. In contrast, the estimated numbers of granulosa cells expressing *Hsd3b1*, *Hsd17b1*, and *Cyp19a1* follow a fluctuating pattern, decreasing from 2 to 4 wk before rising again at 5 wk. Notably, 2 wk corresponds to the time of mini-puberty. (**H**) Estimated numbers of granulosa cells in follicles across developmental subclasses: primordial, primary, secondary, antral, remodeling, and mitotic cycling.

The online version of this article includes the following figure supplement(s) for figure 6:

**Figure supplement 1.** Analysis of single-cell RNA sequencing.

**Figure supplement 2.** Analysis of other cell types.

**Figure supplement 3.** Hormone receptor expression and XO mouse line.

the numbers indicate $\log_{10}$ $q$ values for gc0 and gc6, respectively. These findings suggest that granulosa cells in remodeling wave 1 follicles activate specific genes and pathways to stimulate vascular development, ultimately promoting granulosa cell turnover at 4 wk.

## Analyzing mesenchymal and steroidogenic gene expression

We also re-clustered the mesenchymal cells, including theca cells and stromal cells, yielding 21 new clusters to pursue a higher-resolution picture of theca expansion (*Figure 6E*). The re-clustered mesenchymal cells fell into nine groups: late-expanding theca cells (g9), early theca cells (g2 and g11), *Tagln*-expressing fibroblasts (g1, g8, g14, and g17), *Col14a1*-expressing fibroblasts (g10 and g19), *Cd24a*-expressing fibroblasts (g4), *Cxcl12*-expressing fibroblasts (g3, g5, and g20), *Hoxd9*-expressing fibroblasts (g0), *Col11a1*-expressing fibroblasts (g6 and g7), and mitotic cells expressing *Mki67* (g12, g13, g15, and g18) (*Figure 6F*, *Figure 6—figure supplement 1D*). The expression profile of cluster g16 remains unclear, and it has been provisionally labeled 'TBD'. Theca cells in g9 highly expressed steroidogenic genes such as *Star*, *Cyp11a1*, *Hsd3b1*, and *Cyp17a1* throughout 2–6 wk and peaked in expression and cell number at 5 wk. In contrast, theca cells in cluster g11 exhibited relatively low levels of steroidogenic enzymes and stable cell numbers, except at 5 wk when levels rose sharply to reach 20–50% of those in g9. Both groups expressed the LH receptor (*Lhcgr*) and prostaglandin E synthase 2 (*Ptges2*), previously implicated in theca cell androgen production (*Erickson et al., 1985*). However, *Lhcgr* levels in g11 started lower and increased more dramatically at 5 wk than in g9. Further analysis of other cell types was also performed (*Figure 6—figure supplement 2*).

Theca cells are not genetically capable of converting their androgen products into estrogen and instead rely on exporting them to granulosa cells for this purpose. *Star*, a crucial gene responsible for transporting cholesterol from the cytoplasm into mitochondria, is essential for steroidogenesis. The percentage of steroidogenic theca cells and interstitial gland cells expressing *Star* increased from approximately 50% to over 80% at 5 wk (*Figure 6G*). Our findings demonstrate that steroidogenic theca cells and interstitial gland cells increase in number and acquire a full complement of androgen biosynthetic enzymes during peri-puberty. In contrast, granulosa cells expressing *Hsd3b1*, *Hsd17b1*, and *Cyp19a1*, indicating the potential to synthesize estrogen, peaked at 6 wk (*Figure 6H*), consistent with estrogen levels lagging behind androgen levels during puberty (*Galas et al., 2012*; *Trova et al., 2021*). Moreover, no meaningful expression level of follicle-stimulating hormone receptor (*Fshr*) was observed across cell types and samples (*Figure 6—figure supplement 3A*).

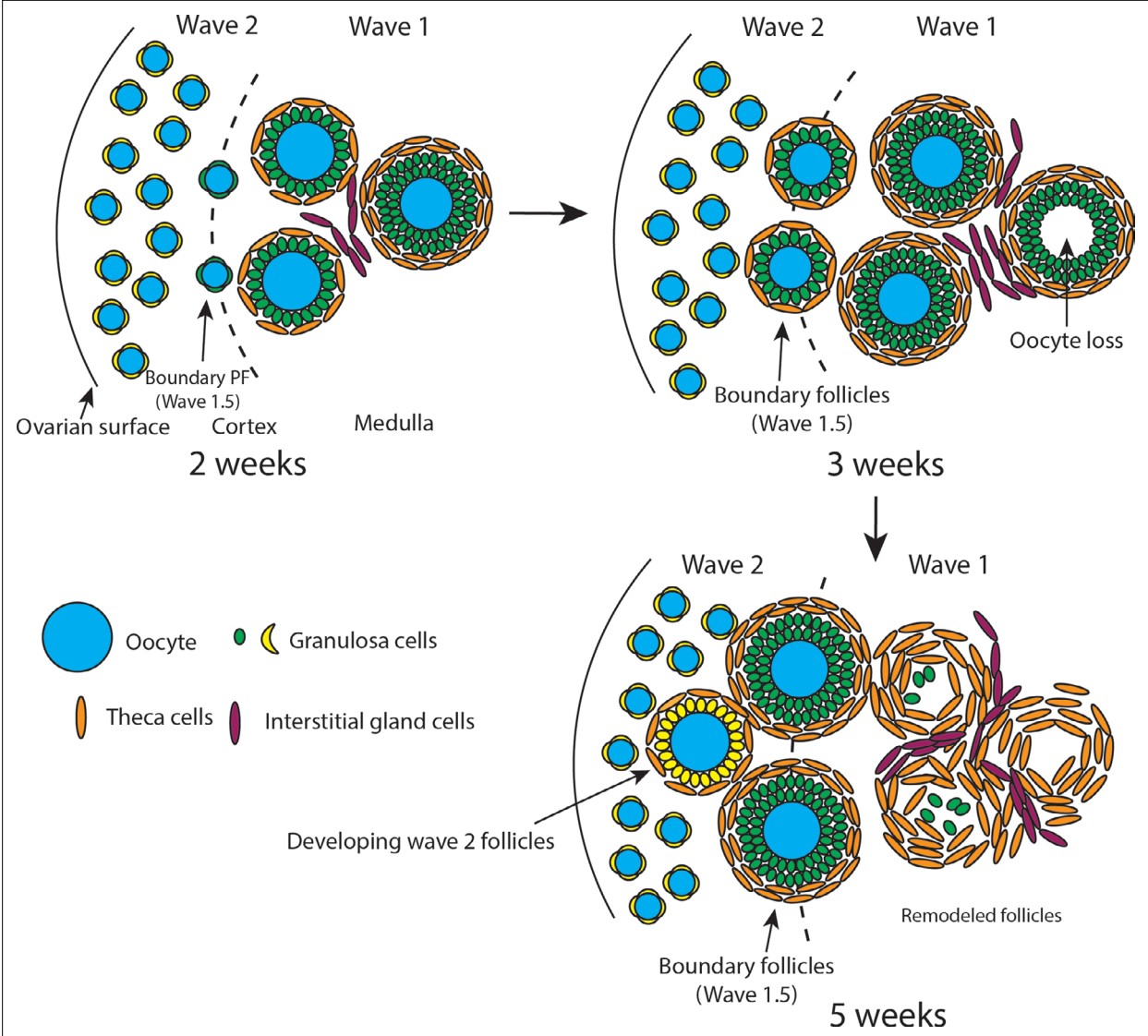

**Figure 7.** Model of wave 1 follicle development and function. Wave 1 follicles undergo remodeling during the peri-pubertal period, expanding the population of androgen-producing theca and interstitial gland cells. Gradients between the medulla and cortex facilitate the sequential activation of primordial follicles, a process that continues into early adulthood. We refer to these sequentially activated follicles as boundary follicles (wave 1.5). These follicles, whose granulosa cells are derived from bipotential pre-granulosa (BPG) cells, remain dormant in the medulla and cortex until recruiting signals—either positive or negative—become available. Boundary follicles are typically recruited around 2 weeks of age and constitute the majority of developing follicles by 5 weeks. In parallel, wave 1 follicles remodel to support androgen production through their remaining theca cells and associated interstitial gland cells.

## Discussion

### Wave 1 follicular remodeling generates a cluster of androgen-producing cells in the juvenile mouse ovary

Our work characterizes in detail the development of the first wave of mouse ovarian follicles at the cellular and gene expression levels. In agreement with earlier studies (*Hirshfield and DeSanti, 1995*; *Gougeon, 1996*; *McGee and Hsueh, 2000*), 80–100% of these follicles turn over rather than generating fertilizable oocytes. We document the cellular and genetic changes within each major follicular cell type and detail the complex events that ultimately produce a large cluster of amplified steroidogenic theca cells and interstitial gland cells, along with a residue of granulosa cells, within the ovarian medulla by 5 wk. This process was previously termed wave 1 follicle atresia and was known to be associated with increased steroidogenesis and vascularization (*Magoffin, 2005*). Our analysis

characterizes the cellular and molecular programs in each wave 1 follicular cell type that cause them to turn over, remodel, change in number, associate with new mesenchymal cells, and modulate their gene expression to produce steroid hormones from the ovary in a timely fashion during juvenile development. Our findings show that wave 1 follicles play a role in steroid production during juvenile development, especially at puberty.

## Wave 1 follicles give rise to few mature oocytes that generate offspring

Boundary follicles, as well as wave 1 follicles, were labeled in our *Foxl2* lineage-tracing experiments and in early experiments of *Zheng et al., 2014a*. Boundary primordial follicles activate as early as 2 wk and produce most of the first follicles to develop fully, ovulate, and generate offspring (*Figure 3*). It remains possible that a small number of wave 1 follicles, approximately 26 or fewer, do develop to maturity. However, the contribution of wave 1 follicles to fertility is, at most, minor. Thus, early-activated wave 2 follicles at the medullary–cortical boundary, rather than wave 1 follicles, are largely responsible for early fertility. Most non-boundary wave 2 follicles only begin to activate between puberty and sexual maturity, possibly in concert with the onset of pulsatile hypothalamic–pituitary–gonadal axis activity.

## Wave 1 follicles may be a conserved feature in other animals, including humans

The developing ovaries of many species produce follicles that do not complete oogenesis or ovulation (*DeFalco and Capel, 2009*; *Jiménez, 2009*). For example, an early wave of germ cells in *C. elegans* hermaphrodites develops into sperm, while during adult life only oocytes are generated (*Pazdernik and Schedl, 2013*). An initial wave of germ cells in the mouse testis develops during juvenile stages but later breaks back down into individual cells, possibly to expand the stem cell pool (*Lei and Spradling, 2013*). Primordial germ cells in *Drosophila* pupal ovaries develop directly into the first ovulated oocytes, while all subsequent oocytes derive from germline stem cells that provide lifelong fertility (*Zhu and Xie, 2003*).

Human follicles begin developing, as in mice, long before there is any chance of ovulation. A fraction of primordial human follicles starts to form during fetal stages, and 4–6% have progressed to the primary follicle stage prior to birth (*Forabosco and Sforza, 2007*). Such early-developing follicles have no prospect of surviving the 8–13 more years until the onset of puberty when they might contribute to fertility. Juvenile human ovaries aged 3 months to 8 years also contain 2–13% of small growing follicles that had diverged from a normal developmental pattern. 50–60% of large follicles showed signs of atresia, including loss of oocytes and most granulosa cells, but retention of some theca cells (*Himelstein-Braw et al., 1976*). Currently, there is little understanding of the developmental mechanisms or logic underlying human fetal and juvenile follicle production. Early follicles in humans might undergo atresia and remodeling to enhance production of androgens or other steroid hormones to influence organ and brain development (*Cui et al., 2013*; *Bell, 2018*). Studies on the development of mouse wave 1 follicles and their component cell types should aid in understanding early follicular waves in many species.

## Identification of the earliest activating wave 2 follicles at the medullary–cortical boundary

A long-standing question in mammalian reproduction has been the activation order of quiescent wave 2 primordial follicles during the life cycle (*McGee and Hsueh, 2000*). In addition to over 200 wave 1 follicles per ovary in the medulla that begin rapid growth at birth, we identified more than 400 wave 2 primordial follicles near the medullary–cortical boundary. Their granulosa cells in 2-wk ovaries were also labeled in a mosaic fashion with *Foxl2-CreERT2* activated at E16.5 or even earlier. Boundary follicles appear to correspond largely or entirely to the primordial follicle subclass poised for early activation and located at the medullary–cortical boundary (*Meinsohn et al., 2021a*; *Meinsohn et al., 2021b*). Our data show that this substantial class of early primordial follicles does begin developing shortly after 2 wk, giving them sufficient time to reach the secondary and antral stages at 5 wk. It remains possible that a small number of wave 1 follicles, approximately 26 or fewer, do develop to

maturity. However, it seems clear that the great majority of early progeny derive from boundary or other early-developing wave 2 follicles, rather than wave 1 follicles that escape atresia.

## Mouse follicle waves express distinct genes

We found that differences in gene expression between wave 1 and 2 granulosa cells exist and may contribute to, or even control, the distinct fates of mouse follicular waves. Previously, gene expression differences had been mapped beginning as early as E14.5, and many were characterized between E16.5 and P5 (*Niu and Spradling, 2020*). Those studies suggested that wave 1 follicles in the medulla differentially express several genes involved in steroid hormone production, while wave 2 granulosa cells in the cortex express genes that might predispose follicles to enter quiescence. This study extends knowledge of differences in gene expression within follicular waves during juvenile development up to 6 wk. Wave 1 follicles express much higher levels of *Wnt4* and *Igfbp5* than wave 2 follicles. Wnt4 has been genetically linked in humans to Müllerian dysgenesis and hyperandrogenism (*Choussein et al., 2017*), perhaps reflecting its known roles in promoting female versus male development. High *Wnt4* expression in wave 1 granulosa cells may indicate that *Wnt4*-mediated downregulation of androgen production at 2 wk becomes reduced by granulosa cell degradation, promoting the subsequent upregulation of androgen levels at puberty. *Igfbp5*, the most conserved of the IGFBP family, is also a potent regulator of female reproductive tissue (*Schneider et al., 2002*). Thus, while changing levels of gonadotropins in young juvenile mice likely play a role in wave 1 atresia (*Edson et al., 2009*), intrinsic differences in gene expression within different follicular waves at the same stage exist and have the potential to be important.

## The timing of follicular activation begins to be specified early in ovarian development

Finding that a subset of primordial follicles located near the medullary–cortical boundary comprises many of the earliest activated wave 2 follicles raises the question of how early activation is controlled. It is well established that upregulation of insulin-mediated growth pathways accompanies follicle activation (*Edson et al., 2009*). Boundary follicles may become poised to activate before most other wave 2 follicles because their proximity to the nutrient-rich medulla enhances growth signaling, weakens quiescence, and promotes activation of *Nr5a2* and *Foxl2* expression, genes expressed in granulosa cells of all growing follicles. Wave 1 granulosa cells in the medulla differentially express growth-promoting genes such as *Igfbp5*, *Grb14*, and *Ghr*, as well as signaling genes like *Itga6* and *Inha*, relative to wave 2 follicles growing during 4 and 5 wk.

However, evidence increasingly suggests that follicular waves do not simply depend on differences in their local microenvironments. These follicle groups form at times in fetal ovary development that overlap with other critical events of gonad development. The progenitors of bipotential reproductive cell types like BPGs and mesenchymal steroidogenic precursors that arise early in both male and female gonad development remain elusive (*Edson et al., 2009*). These cells include stem-like populations that later give rise to theca cells and interstitial gland cells. We were able to preferentially mark both wave 1 follicles and boundary follicles using *Foxl2-CreERT2* as early as E14.5, suggesting that they are specified as early as bipotential and later progenitors and involve intercellular signaling events linked to sex determination. Progress in answering basic questions of gonad development would undoubtedly advance our understanding of how wave 1 and boundary follicles are specified.

## Genetic studies of wave 1 follicles

We explored genotypes that reduce the number of wave 1 follicles in order to look for the genetic consequences of decreased wave 1 activity. In Turner syndrome, wave 1 follicle development is reported to be reduced significantly (*Miura et al., 2017*). This imbalance might lower androgen levels, potentially affecting juvenile development. Consistent with hormonal disruption, hormone replacement therapy significantly improves the development of individuals with Turner syndrome (*Klein et al., 2018*). Moreover, androgens are known to play an important role in this effect (*Zuckerman-Levin et al., 2009*; *Viuff et al., 2022*).

We examined XO mice and looked for changes in wave 1 follicle numbers (*Figure 6—figure supplement 3B, C*). Activated follicle numbers were reduced compared to XX controls at 1 wk. However, the reduction was transient, and by 2 wk, prior to remodeling and theca cell expansion, XO and XX

females had similar numbers of activated follicles. Consequently, we could not draw any conclusions regarding the functional role of wave 1 follicles and their remodeling from these studies.

## Steroid hormones mediate sexual development in juvenile mice and during peri-puberty

Steroid hormones influence the development of sexually dimorphic organs, including the brain (*Simerly, 2002*; *Arnold, 2009*). They have multiple sources, including steroids in the maternal circulation during pregnancy, as well as subsequent systemic and local steroid production during juvenile stages. Wave 1-derived theca cell numbers are high during 4 and 5 wk of juvenile development, a period known as peri-puberty that has been associated with sexual behavior development and with high levels of androgens, as well as relatively low estrogen production (*François et al., 2017*; *Trova et al., 2021*; *Devillers et al., 2023*). While the sensitivity of the brain to steroid influences is probably declining during this period, sexually dimorphic circuits are still being shaped in ways that are important for sexually dimorphic behaviors in female and male rodents (*Sisk and Zehr, 2005*; *Templin et al., 2019*; *Trova et al., 2021*; *Yoest et al., 2023*). Potential targets of this influence include the anteroventral periventricular nucleus, which is larger in female than male rodents and helps regulate female reproductive functions such as ovulation and gonadotropin production. Sex hormones produced in females during puberty also enhance cell death in the primary visual cortex, reducing its volume in females compared to males. Thus, wave 1-generated steroid production may influence sexual development.

## Wave 1 follicle remodeling, polycystic ovarian syndrome, and human follicular waves

The work reported here may be relevant to the origin and high frequency of polycystic ovary syndrome (PCOS) (*McCartney and Marshall, 2016*). A possible connection between follicular atresia, androgen excess, and PCOS has been discussed previously (*Erickson et al., 1985*; *Magoffin, 2005*). Interestingly, some histological characteristics of remodeling mouse wave 1 follicles resemble those of follicles within the ovaries of individuals with PCOS (*Chang, 2007*; *Franks and Hardy, 2010*; *Walters et al., 2019*). A good place to start toward further progress on this connection would be to characterize the development and gene expression of early human follicles and compare them to findings reported in other systems, including mice.

## Materials and methods
### Antibodies

| Name | Source | Identifier | Dilution |
|---|---|---|---|
| Anti-EGFP (Chicken) | Aves labs | GFP-1020 | 1:300 |
| Anti-DDX4 (Rabbit) | Abcam | ab13840 | 1:300 |
| Anti-MSY2 (Mouse) | Abcam | sc-393840 | 1:100 |
| Anti-tdTomato (Goat) | Origene Tech | AB8181-200 | 1:300 |
| Anti-HSD3B1 (Mouse) | Santa Cruz | sc-515120 | 1:100 |
| Anti-CYP17A1 (Rabbit) | Cell Signaling Technology | 94004T | 1:100 |
| Anti-PLIN1 (Rabbit) | Abcam | ab3526 | 1:100 |
| DAPI | Sigma | D9542-10mg | 0.5 µg/ml |
| Alexa Fluor 488 AffiniPure Donkey Anti-Chicken IgY (IgG) (H+L) | Jackson ImmunoResearch | 703-545-155 | 1:300 |

*Continued on next page*

*Continued*

| Name | Source | Identifier | Dilution |
|---|---|---|---|
| Alexa Fluor 594 AffiniPure Donkey Anti-Goat IgG (H+L) | Jackson ImmunoResearch | 705-585-147 | 1:300 |
| Alexa Fluor 488 donkey anti-mouse IgG | Thermo Fisher Scientific | A21202 | 1:300 |
| Alexa Fluor 568 donkey anti-mouse IgG | Thermo Fisher Scientific | A10037 | 1:300 |
| Alexa Fluor 488 donkey anti-rabbit IgG | Thermo Fisher Scientific | A21206 | 1:300 |
| Alexa Fluor 568 donkey anti-rabbit IgG | Thermo Fisher Scientific | A10042 | 1:300 |
| Alexa Fluor 647 donkey anti-rabbit IgG | Thermo Fisher Scientific | A31573 | 1:300 |

## Primers

| Name | Forward (5′–3′) | Reverse (5′–3′) | Size (bp) | Chromosome | Note |
|---|---|---|---|---|---|
| XO-qPCR-Tg | GACTAGGTTCATAGGCACTGG | CCGCCAAAACTCCTTCTCTAC | 123 | X | Genotyping for XO mice |
| XO-qPCR-Tg-Probe | [5HEX]CCCCAGATGGTACCCACAGAACTTG | | | X | |
| XO-qPCR-IPC | CACGTGGGCTCCAGCATT | TCACCAGTCATTTCTGCCTTTG | 74 | 12 | |
| XO-qPCR-IPC-Probe | [Cy5]CCAATGGTCGGGCACTGCTCAA | | | 12 | |
| Foxl2-Mutant | AGAGAAGAGAGTGAGAGCCG | GTCCAGCTCGACCAGGATGG | 221 | 9 | Genotyping for *Foxl2-CreERT2* |
| Foxl2-WT | AGAGAAGAGAGTGAGAGCCG | GAGCGCCACGTACGAGTACG | 335 | 9 | |
| Cyp17a1-Tg | GCTGTAGCTTCTCCACTCCAC | CAGGTTTTGGTGCACAGTCA | 120 | | Genotyping for *Cyp17a1-iCre* |
| Cyp17a1-IPC | AGTGGCCTCTTCCAGAAATG | TGCGACTGTGTCTGATTTCC | 521 | | |

## Experimental mice

All animal experiments were conducted in compliance with the Institutional Animal Care and Use Committee guidelines of the Carnegie Institution of Washington (CIW). The mice were housed in specific pathogen-free facilities at CIW. The strains of mice used in the study were acquired from Jackson Laboratory, including C57BL/6J (strain #: 000664), DBA/2J (strain #: 000671), B6;129P2-*Foxl2tm1(GFP/cre/ERT2)Pzg*/J (*Foxl2-CreERT2*, strain #: 015854), STOCK XO/J (XO, strain #: 036414), B6CBACaF1/J-*Aw-J*/A (strain #: 001201), and B6;SJL-Tg(Cyp17a1-icre)AJako/J (*Cyp17a1-iCre*, strain #: 028547). The *Rosa26-LSL-EYFP* reporter mice, B6.129X1-*Gt(ROSA)26Sortm1(EYFP)Cos*/J (strain #: 006148), were acquired from Jackson Laboratory and backcrossed onto a 129S1/SvImJ (129S1, strain #: 002448) background in our lab. The *Rosa26-LSL-tdTomato* reporter mice were generously provided by Chen-Ming Fan at the CIW. CD-1 IGS mice (strain #: 022) were obtained from Charles River Laboratories.

## Lineage tracing experiments

To visualize the expression patterns of *Foxl2* and *Cyp17a1*, we generated reporter mouse lines by crossing *Rosa26-EYFP* or *Rosa26-tdTomato* mice with *Foxl2-CreERT2* or *Cyp17a1-iCre* (**Zhou et al., 2022**) mice. The presence of a vaginal plug was designated as embryonic day 0.5 (E0.5). Pregnant females were injected intraperitoneally (IP) with 1 mg TAM (MilliporeSigma, #T5648; 10 mg/mL dissolved in corn oil) per 35 g body weight at E16.5 or E14.5 (*Foxl2-CreERT2*). Pups were either delivered naturally or, if not born on the expected delivery day, delivered by C-section and fostered

by a CD1 female mouse. The day of birth was considered P0. Gonads were collected from fetuses, or ovaries from pups, at the indicated stages for analysis. The loss of one *Foxl2* allele in *Foxl2-CreERT2* mice, along with TAM administration, does not significantly impact ovarian development or overall mouse health.

To investigate the expression of *Foxl2* in remodeled follicles, TAM was injected IP at a concentration of 1 mg per 35 g body weight into female mice at 5 weeks of age. The ovaries were dissected one week later and subjected to whole-mount staining for analysis.

All experiments were conducted with at least three biological replicates from two litters and were repeated twice.

## Immunostaining

Whole-mount staining was performed as previously described (*Li et al., 2017*). Briefly, gonads or ovaries were fixed in 4% paraformaldehyde (PFA) overnight at 4°C. The tissues were then washed three times in washing buffer (0.3% Triton X-100 and 0.5% thioglycerol in PBS) and incubated in blocking buffer (1% normal donkey serum, 1% BSA, and 0.3% Triton X-100 in PBS) overnight at 37°C. The samples were incubated with primary antibodies in dilution buffer (1% BSA and 0.3% Triton X-100 in PBS) for 2 days at 37°C. After washing in washing buffer overnight at 37°C, the samples were incubated with secondary antibodies in dilution buffer for another 2 days at 37°C, followed by washing in washing buffer with DAPI overnight at 37°C. To facilitate whole-tissue imaging, the clearing buffer $C_e3D$ (prepared by dissolving 4 g Histodenz in 2.75 ml of 40% [vol/vol] *N*-methylacetamide) was applied for one additional day.

For whole-ovary imaging, ovaries in clearing buffer $C_e3D$ were placed in glass-bottom MatTek dishes (MatTek, P35G-0-14-C) and imaged using a Leica TCS SP8 confocal microscope or a Leica Stellaris 8 DIVE confocal microscope. Images were collected every 3–10 μm. Data were analyzed using Fiji or Imaris. Follicle quantification was performed with Imaris by manually labeling follicles using the spot model. Data were collected from at least two litters.

For fluorescence cryosection staining, dissected tissues were fixed in 4% PFA overnight at 4°C. After washing three times in PBS, the tissues were dehydrated sequentially in 10% and 30% sucrose solutions and embedded in O.C.T. compound for cryosectioning. Sections were cut at 5 μm thickness and stored at –20°C until staining. After antigen retrieval in 0.01% sodium citrate buffer at boiling temperature for 20 min, slides were sequentially blocked for 1 hr in staining buffer (5% normal donkey serum, 3% BSA, and 0.1% Tween-20 in PBS) at room temperature, incubated with primary antibodies in staining buffer overnight at 4°C, and then with secondary antibodies for 1 hr at room temperature. This was followed by counterstaining with DAPI for 10 min before mounting. The samples were then imaged with an upright SP5 microscope, and the images were analyzed using Fiji.

## Oil Red O staining

Oil Red O staining was performed using the Oil Red O Stain Kit (Abcam, ab150678) according to the manufacturer's protocol. Briefly, the slides were placed in propylene glycol for 5 min at room temperature, followed by incubation in Oil Red O solution at 60°C for 10 min. The slides were then differentiated in 85% propylene glycol for 1 min and rinsed twice in distilled water. Next, the slides were incubated in hematoxylin for 1 min, rinsed thoroughly in tap water, and again twice in distilled water. After applying an aqueous mounting medium and coverslip, the slides were imaged using a Nikon Eclipse E800 microscope.

## Genotyping

Tissues were dissected and incubated in lysis buffer (30 mM Tris-HCl, 0.5% Triton X-100, 200 μg/ml Proteinase K) at 55°C overnight in a water bath. The lysates were then inactivated by heating at 95°C for 5 min.

For conventional PCR, 1 μl of the lysate was added to the PCR mixture (primer sequences are provided in the supplemental data). For quantitative PCR (qPCR) to genotype XO mice, primers and probes (sequences included in the supplemental data) were designed to amplify and detect target DNA sequences from both the X chromosome and an autosome. qPCR was performed using a CFX96 Real-Time PCR System. The amplification cycle threshold (Ct) of the X chromosome was compared

with that of the autosome to determine the genotype, distinguishing between XO (one X chromosome) and XX (two X chromosomes) females.

## Single-cell RNA sequencing

Mouse ovaries were first dissected in PBS and dissociated using 0.25% Trypsin/EDTA (Thermo Fisher, #25200056) at 37°C for approximately 20 min. Dissociation was halted by adding FBS (Thermo Fisher, #10439016) to a final concentration of 10%. The resulting cell suspension was filtered through a 100 µm strainer (Fisher, #22-363-549) to remove undissociated structures. Cells were pelleted by centrifugation at $300 \times g$ for 3 min using an Allegra X-14R centrifuge (Beckman Coulter) and resuspended in PBS containing 0.04% BSA (PBSB). To further eliminate cell clumps, the suspension was filtered through a 40-µm strainer (Fisher, #22-363-547), followed by centrifugation at $300 \times g$ for 5 min. The cells were then resuspended in PBSB and subjected to FACS sorting (BD FACSAria III Cell Sorter) to collect live cells, using SYTOX Orange Dead Cell Stain (Thermo Fisher, S34861) for viability selection. After a final centrifugation at $300 \times g$ for 5 min, the cells were resuspended in PBSB at a concentration of 1000 cells/µl before proceeding with single-cell sequencing.

Single-cell libraries were prepared using the Chromium Next GEM Single Cell 3′ Reagent Kits v3.1 (PN-1000128, 10x Genomics), with the goal of capturing 10,000 cells per sample. Sequencing was performed on the Illumina NextSeq 500 platform according to 10x Genomics recommendations: paired-end, single-index reads with Read 1 = 28 cycles, Index 1 = 8 cycles, and Read 2 = 91 cycles. Sequencing depth was ~50,000 reads per cell. Raw sequencing data were processed using the Cell Ranger pipeline (v3.1.0 and v6.0.1) with default settings.

Further analysis was performed using the Seurat package (v4.1.0). Count data were imported with the 'Read10X()' and converted into a Seurat object using 'CreateSeuratObject()'. The libraries (2, 3, 4, 5, and 6 wk) were preprocessed with 'NormalizeData()', followed by 'FindVariableFeatures()', 'FindIntegrationAnchors()', and 'IntegrateData()' using default settings. The integrated object was further processed with 'ScaleData()', 'RunPCA(npcs = 50)', 'RunUMAP(dims = 1:30)', 'FindNeighbors(dims = 1:30)', 'FindClusters(resolution = 1.0)', and 'RunTSNE(dims = 1:30)' with default parameters to determine cell clusters. Granulosa cells and mesenchymal cells were further analyzed by subsetting with 'subset()', followed by the same clustering workflow described above.

## Electron microscopy

Mouse ovaries were excised and fixed in 2% PFA and 2.5% glutaraldehyde in 0.1 M PIPES buffer (MilliporeSigma, #P6757, pH 7.4) at 4°C overnight. The ovaries were then washed and treated with 50 mM glycine in 0.1 M PIPES buffer for 15 min, followed by washing with 0.1 M PIPES buffer. Tissue pieces were then postfixed with 1% osmium tetroxide and 1.5% potassium ferrocyanide in 0.1 M PIPES buffer for 1 hr, washed with water, and stained en bloc with 1% (wt/vol) uranyl acetate for 1 hr.

The ovaries were serially dehydrated in graded ethanol solutions (30%, 50%, 75%, 85%, 95%, and 100%), followed by two exchanges of 100% acetone. After dehydration, the ovaries were infiltrated and embedded in EMbed 812 epoxy resin (Electron Microscopy Sciences) according to the manufacturer's recommendations. Ultrathin sections (~70 nm) were cut on a Leica EM UC7 ultramicrotome and examined using a Hitachi HT7800 transmission electron microscope operated at 80 kV. Images were acquired using an AMT NanoSprint 12 camera.

## Statistical analysis

Data are reported as means ± SD of biological replicates. No statistical methods were used to predetermine sample size. Differences between two groups were analyzed using a two-tailed Student's $t$-test. A p-value <0.05 was considered statistically significant and marked with one asterisk (*), $p < 0.01$ with two (**), $p < 0.001$ with three (***), and $p < 0.0001$ with four (****).

## Acknowledgements

We are grateful to Allison Pinder and Dr. Fredrick Tan for their assistance with the genomic experiments and analyses. We thank Dr. Mahmud Siddiqi for his help with optical microscopy and image analyses, and Dr. Ru-Ching Hsia for expert electron microscopy. We also thank the Carnegie Embryology support staff and Spradling lab members for their valuable comments on the study, especially Haolong Zhu for assistance with scRNA-seq data analysis.

## Additional information

### Funding

| Funder | Grant reference number | Author |
|---|---|---|
| Howard Hughes Medical Institute | Allan Spradling is an HHMI Investigator | Qi Yin<br>Allan C Spradling |

The funders had no role in study design, data collection, and interpretation, or the decision to submit the work for publication.

### Author contributions

Qi Yin, Conceptualization, Formal analysis, Investigation, Methodology, Writing – original draft, Writing – review and editing; Allan C Spradling, Conceptualization, Resources, Data curation, Formal analysis, Supervision, Funding acquisition, Validation, Writing – original draft, Writing – review and editing

### Author ORCIDs

Allan C Spradling (iD) https://orcid.org/0000-0002-5251-1801

### Ethics

All our work was carried out in the Carnegie Institution mouse facility under the supervision of the Carnegie Institution Animal Care Committee using approved protocols and standards.

Reviewer #1 (Public review): https://doi.org/10.7554/eLife.107352.2.sa1
Reviewer #2 (Public review): https://doi.org/10.7554/eLife.107352.2.sa2
Author response https://doi.org/10.7554/eLife.107352.2.sa3

## Additional files

### Supplementary files

Source data 1. Summary of general information for all five samples (2wk to 6wk). Excel file containing sample metadata, including sample code, number of reads, number of cells, average genes per cell, average UMI per cell, total detected genes, reference transcriptome used for alignment, and sample name.

Source data 2. Average gene expression in 31 clusters (0 to 30) from five samples (2wk to 6w). Excel file with average expression values of genes generated using Seurat. Column A: gene name. Row 3: cluster and sample name (e.g., 0_O2w, where '0' indicates cluster 0 and 'O2w' indicates the 2-wk ovary sample).

Source data 3. Average gene expression of re-clustered granulosa cells and Metascape analysis of selected clusters. Excel file with multiple sheets: (1) Expression_2wk_6wk_Granulosa: average expression values of genes in re-clustered granulosa cells (gc0–gc17) from five samples, generated using Seurat. Column A: gene name. Row 3: cluster and sample name. (2) gc0_Metascape: Gene Ontology (GO) enrichment analysis of cluster gc0 (Metascape). (3) gc6_Metascape: GO enrichment analysis of cluster gc6 (Metascape). (4) gc11_Metascape: GO enrichment analysis of cluster gc11 (Metascape).

Source data 4. Average gene expression of re-clustered mesenchymal cells (g0–g20). Excel file containing average expression values generated using Seurat. Column A: gene name. Row 3: cluster and sample name (e.g., 0_O2w, where '0' indicates cluster 0 and 'O2w' indicates the 2-wk ovary sample).

Source data 5. Top 250 preferentially expressed genes in initial and re-clustered mesenchymal cell clusters. Excel file with three sheets:(1) Top250_Initial: top 250 genes (Column G) in initial clusters c0–c30 (Column F). (2) Top250_Granulosa: top 250 genes (Column G) in re-clustered granulosa cell clusters gc0–gc17 (Column F) . (3) Top250_Mesenchymal: top 250 genes (Column G) in re-clustered mesenchymal cell clusters g0–g20 (Column F) .

Source data 6. Cell numbers in initial and re-clustered clusters from five samples (2wk to 5wk). Excel file with three sheets: (1) Cells_in_initial_clusters: cell numbers in initial clusters (c0–c30) (Column

A). (2) Cells_in_granulosa_recluster: cell numbers in re-clustered granulosa cell clusters (gc0–gc17) (Column A). (3) Cells_in_mesenchymal_recluster: cell numbers in re-clustered mesenchymal cell clusters g0–g20 (Column A). Row 3: sample names.

MDAR checklist

## Data availability

The scRNA-seq data have been deposited in the NIH GEO database under accession number GSE268466.

The following dataset was generated:

| Author(s) | Year | Dataset title | Dataset URL | Database and Identifier |
|---|---|---|---|---|
| Yin Q, Spradling AC | 2025 | Single-cell RNA sequencing of ovarian cells from mice aged 2 weeks, 3 weeks, 4 weeks, 5 weeks, and 6 weeks | http://www.ncbi.nlm.nih.gov/geo/query/acc.cgi?acc=GSE268466 | NCBI Gene Expression Omnibus, GSE268466 |

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
