## [Editor Report · eLife Assessment]

This **important** study reports that two distinct waves of ovarian follicles contribute to oocyte production in mice. The paper provides large amounts of data that will benefit future studies, although the methods and analysis are considered **incomplete** at present. Justification for the criteria of wave 1 follicles would benefit from further explanation and discussion. This work will be of interest to ovarian biologists and physicians working on female infertility.

---

## [Referee Report · Reviewer #1 (Public review)]

Multiple waves of follicles have been proven to exist in multiple species, and different waves of follicles contribute differently to oogenesis and fertility. This work characterizes the wave 1 follicles in mouse comprehensively and compares different waves of follicles regarding their cellular and molecular features. Elegant mouse genetics methods are applied to provide lineage tracing of the wave 1 folliculogenesis, together with sophisticated microscopic imaging and analyses. Single-cell RNA-seq is further applied to profile the molecular features of cells in mouse ovaries from week 2 until week 6. While extensive details about the wave 1 follicles, especially the atresia process, are provided, the authors also identified another group of follicles located in the medullary-cortical boundary, which could also be labeled by the FoxL2-mediated lineage tracing method. The "boundary" or "wave 1.5" follicles are proposed by the authors to be the earliest wave 2 follicles, which contribute to the early fertility of puberty mice, instead of the wave 1 follicles, which undergo atresia with very few oocytes generated. The wave 1 follicle atresia, which degrades most oocytes, on the other hand, expands the number of theca cells and generates the interstitial gland cells in the medulla, where the wave 1 follicles are located. These gland cells likely contribute to the generation of androgen and estrogen, which shape oogenesis and animal development. By comparing scRNA-seq data from cells collected from week 2 until week 6 ovaries, the author profiled the changes in numbers of different cells and identified key genes that differ between wave 1 and wave 2 follicles, which could potentially be another driver of different waves of folliculogenesis. In summary, the authors provide a high amount of new results with good quality that illustrate the molecular and cellular features of different waves of mouse follicles, which could be further reused by other researchers in related fields. The findings related to the boundary follicles could potentially bring many new findings related to oogenesis.

This paper is overall well-written with solid and intriguing conclusions that are well supported. The reviewer only has some minor comments for the authors' consideration that could potentially help with the readability of the paper.

(1) The authors identify the wave 1.5 follicles at the medullary-cortex boundary, which begin to develop shortly after 2 weeks. Since the authors already collected scRNA-seq data from week 2 until week 6, could any special gene expression patterns be identified that make wave 1.5 follicle cells different from wave 1 and wave 2?

(2) Are Figures 1C and 1E Z projections from multiple IF slices? If so, please provide representative IF slice(s) from Figures 1C and 1E and clearly label wave 1 and wave 2 follicles to better illustrate how the wave 1 follicles are clarified and quantified.

(3) For Figure 3D, please also provide an image showing the whole ovary section, like in Figures 3A and 3C, to better illustrate the localization and abundance of different cells.

(4) In Figure 4H, expressions of HSD3B1 and PLIN1 seem to be detected in almost all medulla cells. Does this mean all medulla cells gain gland cell features? Or there is only a subset of the medulla cells that are actively expressing these 2 proteins. Please provide image(s) with higher magnification to show more clearly how the expression of these 2 proteins differs among different cells.

(5) Figure 5: The authors discussed cell number changes for different types of cells from week 2 to week 6. A table, or some plots, visualizing numbers of different cell types, instead of just providing original clusters in Dataset S6, at different time points, would make the changes easier to observe.

(6) Figure S7: It would be more helpful to directly show the number of wave 1 follicles.

(7) Did the fluorescence cryosection staining (Line 587 - 595) use the same buffers as in the whole-mount staining (Line 575 - 586)? Please clarify.

(8) In Line 618, what tissue samples were collected? Please point out clearly.

---

## [Referee Report · Reviewer #2 (Public review)]

Summary:

This study explores an important question concerning the developmental trajectory of wave 1 ovarian follicles, leveraging valuable tools such as lineage tracing and single-cell RNA sequencing. These approaches position the authors well to dissect early follicle dynamics. The study would benefit from more in-depth analysis, including quantification using the lineage-traced ovaries, and comparison of wave 1 and 2 follicular cells per stage within the single cell dataset.

Strengths:

This study aims to address an important question regarding the developmental trajectories of wave 1 ovarian follicles and how they differ from wave 2 follicles that contribute to long-term fertility. This is an important topic, as many studies on ovarian follicle development rely on samples collected at perinatal timepoints in the mouse, which primarily represent wave 1 follicles, to infer later fertility. The research group has the tools and expertise necessary to tackle these questions.

Weaknesses:

Wave 1 follicles are quantified based on the criteria of oocytes larger than 20 µm located within the medullary region, using whole-mount staining. However, the boundary between the medulla and cortex appears somewhat arbitrary. Quantification using FOXL2-lineage-traced ovaries provides a more reliable method for identifying wave 1 follicles. As the developmental trajectory of wave 1 follicles has been well described in Zhang et al. 2013, it would be valuable to provide a more detailed quantification of both labeled and unlabeled follicles by specific follicle stages. In fact, in Zhang et al. 2013, the authors demonstrated that lineage-labeled primordial follicles can be found at the cortex-medulla boundary, suggesting that the observation of labeled "border follicles" is not unexpected. Quantification by follicle stage would provide greater insight into the timing and development of these follicles.

Similarly, the analysis of wave 1 follicle loss should be performed on lineage-traced ovaries using cell death markers to demonstrate the loss of oocytes and granulosa cells, while confirming the preservation of theca and interstitial cells. In particular, granulosa cell loss should be assessed directly with cell death markers in lineage-traced ovaries, rather than from the loss of tamoxifen-labeled cells, as labeling efficiency varies between follicles (Figure 2G).

Single-cell RNA sequencing presents a valuable dataset capturing the development of first-wave follicles. The use of a 40µm cell strainer during cell collection for the 10x platform may explain the exclusion of larger oocytes. However, it is still surprising that no oocytes were captured at all. The central question, how wave 1 follicular cells differ from wave 2 cells, should be investigated in more depth, with results validated on FOXL2-lineage-traced ovaries (i.e., Wnt4 staining in wave 1 antral follicles versus wave 2 using lineage-traced ovaries). This analysis should span all stages of follicle development. It also appears to be a missed opportunity that the single-cell sequencing analysis was not performed on lineage-traced ovaries, which would have enabled more definitive identification of wave 1-derived cells.

Finally, this study does not directly assess fertility outcomes and should therefore refrain from drawing conclusions about the fertility potential of wave 1 follicles.

---

## [Author Response]

The eLife assessment states that our manuscript is important only as a source of data for others to use in the future. Our methods and analysis of wave 1 follicles were said to be "incomplete" because one of two reviewers claimed we did not prove that 80% of wave 1 oocytes turn over by 5 wk.

We believe that this assessment is simply wrong because critical supporting data already present in the existing manuscript were not understood by one reviewer. Wave 1 follicular oocyte turnover was said to be unproven and to remain uncertain because evidence of death was based only on a lack of Ddx4 staining. New experiments documenting expression of cell death markers, were said to be needed to show the oocytes died. However, our work was not based on the analysis of sectioned material, but used whole mount 3D reconstruction microscopy of cleared ovary preparations. Oocyte death was determined by the absence of an oocyte in fully reconstructed follicles and its replacement with an empty cavity, not just the absence of antibody staining. We included images and complete 3D reconstruction movies documenting these methods. The paper also documents that the holes frequently still contained zona pellucida remnants indicating the former presence of an oocyte. Moreover, we observed many intermediates of oocyte death- shrunken and deformed oocytes- and deformations of follicle structures due to the presence of the empty cavities. Controls showed that Ddx4 staining in the context of 3D imaging always revealed an obvious giant labeled oocyte in 100% of wave 1 follicles prior to death, and in wave 1.5 and wave 2 follicles at all stages. Thus, our methodology is already fully reliable. The reviewer is correct that the entire program of wave 1 development including their programmed turnover would be interesting to explore further. We already provided a large amount of new gene expression information, and documented the first examples of wave 1-specific gene expression. Further studies are not needed for the major conclusions of the paper and can wait for a follow up study.

Secondly, the existence of wave 1.5 is not "speculative," as stated by the reviewing editor. We extensively validated and quantified the existence of wave 1.5 primordial follicles following Foxl2-cre activation at E16.5, and analysis at 2 wks in multiple experiments. Additionally, we showed wave 1.5 follicles were present at the medullar/cortex border at 2 wks even after activation of Foxl2-cre at E14.5. Our paper also connected for the first time wave 1.5 follicles to a population of non-growing, "poised" primordial follicles at this identical location near the medulla/cortex boundary by Meinsohn et al. in 2021. These follicles had not started to develop yet, and their ultimate fate was not known. We followed the development of these follicles and determined several differences in wave 1.5 follicle gene expression compared to wave 1. As noted in the assessment, our findings on wave 1.5 are now already being extended to other systems such as primate ovaries (adopting our name "wave 1.5" from our bioRxiv manuscript). The simultaneous claims that our discovery of wave 1.5 exists is speculative, and also that other people are now finding wave 1.5 follicles in the species they are studying are logically incompatible.

**Response to reviewer 2:**

Line 239-245: Please note that Zhang et al. 2013 also show that lineage-labeled primordial follicles can be found at the cortex-medulla boundary (see their Figure 1B).

The single image in the Zheng et al. 2014 paper may or may not show mosaic primordial follicles, but it would not be surprising since the experiment was identical to experiments in the paper. However, that single picture is only meaningful in the context of our subsequent work reported in the current manuscript. There was no mention of these follicles in the text of Zheng et al. 2014, no documentation or quantitation of their numbers, and no discussion or understanding of their significance. The incorrect conclusions of the paper were that wave 1 follicles- meaning rapidly developing follicles in the medulla- give rise to most early offspring. This conclusion reversed the previously accepted (and essentially correct) view that wave 1 follicles did not contribute significantly to fertility.

"Finally, this study does not directly assess fertility outcomes and should therefore refrain from drawing conclusions about the fertility potential of wave 1 follicles."

We showed by lineage marking that only about 25 of about 200 wave 1 follicles survive even to wk 5. This clearly does prove our conclusion that the great majority of wave 1 follicles do not contribute to fertility.